# Unsupervised Diffusion Solver for Combinatorial Optimization via Combinatorial Adjoint Matching

**Shengyu Feng** [1]  **Tarun Suresh** [2]  **Yiming Yang** [1]

## Abstract

Diffusion-based neural solvers have shown strong promise for combinatorial optimization (CO), but existing methods typically rely on supervised training with large collections of near-optimal solutions. In this work, we extend adjoint-based trajectory optimization methods to discrete combinatorial domains. We formulate diffusion-based CO as a stochastic control problem over Continuous-Time Markov Chains and introduce discrete adjoint dynamics for propagating optimization signals through discrete generative trajectories. Building on this formulation, we propose *Combinatorial Adjoint Matching (CAM)*, an unsupervised training framework for discrete diffusion solvers with structured and low-variance trajectory-level optimization signals. Empirically, CAM consistently outperforms existing unsupervised diffusion baselines and achieves performance competitive with strong supervised diffusion solvers and even traditional solvers across diverse combinatorial optimization problems. Our code is available at https://github.com/Shengyu-Feng/CAM.

## 1. Introduction

Combinatorial optimization (CO) aims to find optimal solutions over discrete and structured decision spaces and is a central topic in computer science, with applications including scheduling, routing, and network design (Korte & Vygen, 2012). Owing to their discrete and highly non-convex nature, many CO problems lack efficient exact solvers, and state-of-the-art (SOTA) performance often relies on carefully engineered heuristics that require substantial domain knowledge and manual effort.

Recent progress in diffusion models (Ho et al., 2020) has led to promising advances in machine-learning-based approaches for CO (Sun & Yang, 2023; Li et al., 2023; Feng et al., 2025). In contrast to one-shot generation methods (Joshi et al., 2019), diffusion models construct solutions through a multi-step stochastic process, allowing them to better represent multi-modal distributions and partially smooth the highly non-convex optimization landscape of CO problems. However, most existing diffusion-based CO solvers rely on supervised training with large collections of near-optimal solutions. This requirement substantially limits their applicability, particularly for problem settings where high-quality solutions are expensive or impractical to obtain. Consequently, developing *unsupervised* training methods for diffusion-based CO solvers has become an important research direction.

Several recent works have explored unsupervised diffusion training for CO. DiffUCO (Sanokowski et al., 2024) formulates diffusion training as an entropy-regularized reinforcement learning (RL) problem. While principled, this approach requires propagating terminal rewards backward through intermediate diffusion steps via policy gradients, resulting in high variance and poor scalability. SDDS (Sanokowski et al., 2025) alleviates this issue by converting the RL objective into a supervised diffusion-style loss using importance sampling from the model itself; however, the variance introduced by importance weights still poses significant challenges. More recently, RLNN (Feng & Yang, 2025) proposes a fully local training objective by interpreting diffusion as a regularized Langevin dynamics. Although this formulation improves stability, its local nature limits the model's ability to capture long-term dependencies and may lead to suboptimal global solutions.

In parallel, adjoint methods (Pontryagin et al., 1962) have become a fundamental tool for trajectory optimization in continuous dynamical systems, including diffusion-based generative modeling (Domingo-Enrich et al., 2024; 2025). Existing RL-based approaches primarily rely on propagating scalar rewards through trajectories, often requiring extensive sampling to reduce variance and assess relative action quality. In contrast, adjoint-based methods compute path gradients through backward propagation along the trajec-

---
[1]Language Technologies Institute, Carnegie Mellon University [2]University of Illinois Urbana-Champaign. Correspondence to: Shengyu Feng <shengyuf@cs.cmu.edu>.

*Proceedings of the 43rd International Conference on Machine Learning*, Seoul, South Korea. PMLR 306, 2026. Copyright 2026 by the author(s).

tory, providing low-variance guidance on how intermediate controls should be adjusted to improve the final outcome.

However, extending such trajectory-level optimization principles to discrete combinatorial domains remains highly challenging due to the absence of differentiable state transitions and the fundamentally discrete nature of combinatorial solution spaces. These challenges therefore motivate the following question:

*How can adjoint-based trajectory optimization methods be adapted to discrete generative processes for combinatorial optimization?*

To address this question, we develop a principled framework for adjoint-based optimization in discrete combinatorial domains. We formulate diffusion-based combinatorial optimization as a stochastic optimal control problem over Continuous-Time Markov Chains (CTMCs) (Anderson, 2012), enabling trajectory optimization in discrete state spaces. Building on this formulation, we introduce discrete adjoint dynamics and establish conditions under which trajectory-level optimization signals can be propagated through discrete generative processes. These components together yield **Combinatorial Adjoint Matching (CAM)**, an unsupervised training framework for discrete diffusion solvers with path-gradient optimization signals.

Unlike prior unsupervised diffusion approaches that rely on dense reward estimation or importance-weighted sampling, CAM utilizes informative terminal improvement signals to guide intermediate diffusion dynamics during training. As a result, CAM substantially reduces the amount of environment feedback required during training while improving optimization stability and inference-time scaling behavior. We evaluate CAM on a diverse set of CO problems, including Maximum Independent Set (MIS), Traveling Salesman Problem (TSP), Maximum Cut, and Capacitated Vehicle Routing Problem (CVRP). Across these tasks, CAM consistently outperforms existing unsupervised diffusion baselines and achieves performance competitive with strong supervised diffusion solvers and even traditional heuristic solvers.

Overall, our results highlight adjoint-based trajectory optimization as a promising direction for scalable and data-efficient neural combinatorial optimization.

## 2. Preliminaries

### 2.1. Problem Statement: Combinatorial Optimization

We formulate a combinatorial optimization (CO) problem as the following constrained optimization problem:

$$\min_{\mathbf{x} \in \{0,1\}^N} a(\mathbf{x}) \quad \text{s.t.} \quad b(\mathbf{x}) = 0, \quad (1)$$

where $a(\mathbf{x})$ denotes the objective to be optimized (e.g., the negative set size in Maximum Independent Set), and $b(\mathbf{x}) \geq 0$ measures the degree of constraint violation (e.g., the number of adjacent selected nodes). Without loss of generality, we consider minimization problems; any maximization problem can be converted into this form.

In practice, neural solvers may produce infeasible solutions. We therefore assume the existence of a surrogate objective function $g : \{0,1\}^N \to \mathbb{R}$ that assigns a cost to any candidate solution and is consistent with the original optimization problem in the sense that

$$\mathbf{x}^\star \in \arg\min_{\mathbf{x} \in \{0,1\}^N} g(\mathbf{x})$$
$$\Rightarrow \mathbf{x}^\star \in \arg\min_{\mathbf{x} \in \{0,1\}^N} a(\mathbf{x}) \quad \text{s.t.} \quad b(\mathbf{x}) = 0. \quad (2)$$

For instance, $g$ can be defined as a penalty-based objective $g(\mathbf{x}) := a(\mathbf{x}) + \beta b(\mathbf{x})$ for a sufficiently large $\beta$, or as the evaluation of solution quality after post-processing neural outputs via greedy decoding.

### 2.2. Continuous-Time Markov Chain

In the discrete domain, we model the generative process as a Continuous-Time Markov chain (CTMC) (Anderson, 2012) over a finite state space $\mathcal{X}$ (e.g., $\{0,1\}^N$ for CO). Let $X_t \in \mathcal{X}$ denote the state at time $t \in [0,1]$. The dynamics are characterized by transition rates $u_t(\mathbf{x}', \mathbf{x})$, which specify the infinitesimal transition probability to $\mathbf{x}' \neq \mathbf{x}$ at time $t$:

$$u_t(\mathbf{x}', \mathbf{x}) = \lim_{\Delta t \to 0} \frac{\mathbb{P}(X_{t+\Delta t} = \mathbf{x}' \mid X_t = \mathbf{x})}{\Delta t}, \quad (3)$$

while $u_t(\mathbf{x}, \mathbf{x}) = -\sum_{\mathbf{x}'} u_t(\mathbf{x}', \mathbf{x})$. In this work, we restrict $\mathbf{x}'$ to the 1-Hamming neighborhood of $\mathbf{x}$ and we use $\mathbf{x}^{(i)}$ to represent the state obtained by flipping the $i$-th coordinate of $\mathbf{x}$.

Given an initial distribution and transition rates, the CTMC induces a trajectory distribution over paths $\mathbf{X} = (X_t)_{t \in [0,1]}$. We denote by $p^{\text{base}}(\mathbf{X})$ a reference process (Kappen, 2005), which serves as a prior over trajectories. A controlled process $p^u(\mathbf{X})$ is obtained by modifying these transition rates through a controlled policy $u$, resulting in a new trajectory distribution in the same state space.

### 2.3. Stochastic Optimal Control

Based on the CTMC formulation above, we formulate the stochastic optimal control (SOC) (Hammer, 1958; Fleming & Rishel, 1975; Sethi & Thompson, 2000) as:

$$\min_u \mathbb{E}_{p^u} \left[ g(X_1) \right] + \tau \text{KL}\big(p^u(\mathbf{X}) \,\|\, p^{\text{base}}(\mathbf{X})\big). \quad (4)$$

Here, $g(X_1)$ is the terminal objective as in the CO formulation, and the KL term regularizes the controlled process

towards the base dynamics, weighted by the temperature $\tau$. We denote the corresponding instantaneous KL cost by $c_t(u; \mathbf{x})$, which, for a CTMC, takes the form

$$c_t(u; \mathbf{x}) = \tau \sum_{\mathbf{x}' \neq \mathbf{x}} \Big( u_t(\mathbf{x}', \mathbf{x}) \log \frac{u_t(\mathbf{x}', \mathbf{x})}{u_t^{\text{base}}(\mathbf{x}', \mathbf{x})} \\ - u_t(\mathbf{x}', \mathbf{x}) + u_t^{\text{base}}(\mathbf{x}', \mathbf{x}) \Big). \quad (5)$$

This naturally induces the cost-to-go function at state $\mathbf{x}$ and time $t$ under control $u$ as

$$J_t(u; \mathbf{x}) = \mathbb{E}_{p^u} \Big[ g(X_1) + \int_{s=t}^{1} c_s(u; X_s) ds \Big| X_t = \mathbf{x} \Big], \quad (6)$$

and yields the following local policy-improvement objective (assuming the fixed cost-to-go):

$$\min_{u_t} \sum_{\mathbf{x}' \neq \mathbf{x}} u_t(\mathbf{x}', \mathbf{x}) \big( J_t(u; \mathbf{x}') - J_t(u; \mathbf{x}) \big) + c_t(u; \mathbf{x}). \quad (7)$$

The key challenge therefore lies in estimating $J_t(u; \mathbf{x}') - J_t(u; \mathbf{x})$. In continuous domains, this quantity can be approximated via the first-order expansion:

$$J_t(u; \mathbf{x}') - J_t(u; \mathbf{x}) \approx \nabla J_t(u; \mathbf{x})^\top (\mathbf{x}' - \mathbf{x}). \quad (8)$$

Here, $\nabla J_t(u; \mathbf{x})$ (or an estimator thereof) is often referred to as the *adjoint* (Pontryagin et al., 1962; Domingo-Enrich et al., 2024; 2025), and can be efficiently computed as a path gradient through the trajectory via a backward Ordinary Differential Equation. However, this gradient-based formulation does not directly extend to discrete state spaces, where conventional gradients are not well defined.

# 3. Method

In this section, we formally establish our formulation and method to extend the adjoint method to combinatorial space.

## 3.1. SOC Objective for Combinatorial Optimization

We begin by formulating a SOC objective for combinatorial optimization and study its behavior in the low-temperature regime $\tau \to 0$:

$$\lim_{\tau \to 0} \min_{u} \mathbb{E}_{p^u} [g(X_1)] + \tau \text{KL} \big( p^u(\mathbf{X}) \,\|\, p^{\text{base}}(\mathbf{X}) \big). \quad (9)$$

We assume a homogeneous base process with a constant transition rate to every neighboring state:

$$u_t^{\text{base}}(\mathbf{x}^{(i)}, \mathbf{x}) = \rho_\tau, \quad (10)$$

where $\rho_\tau = \sigma(-\lambda/\tau)$ and $\lambda > 0$ is fixed. Under this choice, we have

$$\lim_{\tau \to 0} c_t(u; \mathbf{x}) = \lambda \sum_{\mathbf{x}' \neq \mathbf{x}} u_t(\mathbf{x}', \mathbf{x}), \quad (11)$$

which penalizes the aggregate transition rate away from the current state. Therefore, Equation 9 can be interpreted as seeking a shortest path to an optimal solution in the low-temperature limit. We obtain the following characterization of the optimal control as $\tau \to 0$.

**Proposition 3.1.** *Let $\mathbf{x}^\star$ denote the closest minimizer to $\mathbf{x}$ under Hamming distance, assumed to be unique. For sufficiently small but fixed $\lambda$, as $\tau \to 0$, the optimal transition rate satisfies*

$$u_t^\star(\mathbf{x}^{(i)}, \mathbf{x}) = \begin{cases} \dfrac{1}{1-t}, & x_i \neq x_i^\star, \\ 0, & x_i = x_i^\star. \end{cases} \quad (12)$$

## 3.2. Discrete Adjoint

We now introduce the *discrete adjoint*, defined as the difference in cost-to-go between two states under a policy $u$:

$$a_t^u(\mathbf{x}'; \mathbf{x}) := J_t(u; \mathbf{x}') - J_t(u; \mathbf{x}). \quad (13)$$

Intuitively, the discrete adjoint measures the relative future cost of transitioning to a counterfactual state $\mathbf{x}'$ compared to remaining at the current state $\mathbf{x}$. By definition, the discrete adjoint satisfies the terminal condition

$$a_1^u(\mathbf{x}'; \mathbf{x}) = g(\mathbf{x}') - g(\mathbf{x}). \quad (14)$$

For intermediate states, the discrete adjoint evolves according to the following backward recursion.

**Proposition 3.2.** *The discrete adjoint satisfies*

$$- \partial_t a_t^u(\mathbf{x}'; \mathbf{x}) = \sum_{\mathbf{y} \neq \mathbf{x}'} u_t(\mathbf{y}, \mathbf{x}') a_t^u(\mathbf{y}; \mathbf{x}') \\ - \sum_{\mathbf{y} \neq \mathbf{x}} u_t(\mathbf{y}, \mathbf{x}) a_t^u(\mathbf{y}; \mathbf{x}) + c_t(u; \mathbf{x}') - c_t(u; \mathbf{x}). \quad (15)$$

## 3.3. Combinatorial Adjoint Matching

We now specialize the discrete adjoint to combinatorial domains. Under the 1-Hamming neighborhood restriction, we consider $\mathbf{x}' = \mathbf{x}^{(i)}$ and restrict the transitions in Equation 15 to single-coordinate flips:

$$- \partial_t a_t^u(\mathbf{x}^{(i)}; \mathbf{x}) = \sum_j u_t(\mathbf{x}^{(ij)}, \mathbf{x}^{(i)}) a_t^u(\mathbf{x}^{(ij)}; \mathbf{x}^{(i)}) \\ - \sum_j u_t(\mathbf{x}^{(j)}, \mathbf{x}) a_t^u(\mathbf{x}^{(j)}; \mathbf{x}) + c_t(u; \mathbf{x}^{(i)}) - c_t(u; \mathbf{x}), \quad (16)$$

where $\mathbf{x}^{(ij)}$ denotes the state obtained by sequentially flipping the $i$-th and $j$-th coordinates of $\mathbf{x}$.

However, estimating the backward recursion, particularly the terms $u_t(\mathbf{x}^{(ij)}, \mathbf{x}^{(i)}) a_t^u(\mathbf{x}^{(ij)}; \mathbf{x}^{(i)})$, requires sampling an additional trajectory starting from $\mathbf{x}^{(i)}$. This prevents

us from exploiting the local information available from a single sampled trajectory (e.g., $X_t = \mathbf{x}$) as in the continuous setting. To address this issue, we further simplify the recursion using a fixed-point condition analogous to that used in Adjoint Matching (Domingo-Enrich et al., 2025).

We first observe that $u_t(\mathbf{x}^{(j)}, \mathbf{x}) = u_t(\mathbf{x}^{(ij)}, \mathbf{x}^{(i)})$ for any $i \neq j$ under the optimal control $u^\star$, as a direct consequence of Proposition 3.1. Under this condition, we have

$$\sum_{j \neq i} u_t(\mathbf{x}^{(ij)}, \mathbf{x}^{(i)}) a_t^u(\mathbf{x}^{(ij)}; \mathbf{x}^{(i)}) - u_t(\mathbf{x}^{(j)}, \mathbf{x}) a_t^u(\mathbf{x}^{(j)}; \mathbf{x})$$

$$\Rightarrow \sum_{j \neq i} u_t(\mathbf{x}^{(j)}, \mathbf{x}) \Big( a_t^u(\mathbf{x}^{(ij)}; \mathbf{x}^{(i)}) - a_t^u(\mathbf{x}^{(j)}; \mathbf{x}) \Big) \quad (17)$$

$$= \sum_{j \neq i} u_t(\mathbf{x}^{(j)}, \mathbf{x}) \Big( a_t^u(\mathbf{x}^{(ij)}; \mathbf{x}^{(j)}) - a_t^u(\mathbf{x}^{(i)}; \mathbf{x}) \Big),$$

where the final equality follows from decomposing the adjoint into differences of cost-to-go values and recombining the resulting terms. Using $u_t(\mathbf{x}, \mathbf{x}) = -\sum_j u_t(\mathbf{x}^{(j)}, \mathbf{x})$, the above expression simplifies to

$$\sum_{\mathbf{y} \neq \mathbf{x}^{(i)}} u_t(\mathbf{y}, \mathbf{x}) a_t^u(\mathbf{y}^{(i)}; \mathbf{y}). \quad (18)$$

Now, we can further eliminate the term $u_t(\mathbf{x}, \mathbf{x}^{(i)})$ using the following proposition.

**Proposition 3.3.** *Under the assumptions of Proposition 3.1, the optimal control $u^\star$ satisfies, for any $t < 1$,*

$$u_t(\mathbf{x}, \mathbf{x}^{(i)}) a_t^u(\mathbf{x}; \mathbf{x}^{(i)}) + c_t(u; \mathbf{x}^{(i)}) - c_t(u; \mathbf{x})$$
$$= -u_t(\mathbf{x}^{(i)}, \mathbf{x}) a_t^u(\mathbf{x}; \mathbf{x}^{(i)}). \quad (19)$$

Combining the above results yields a "lean" adjoint $\tilde{a}$ satisfying the following recursion under optimality:

$$-\partial_t \tilde{a}_t^u(\mathbf{x}^{(i)}; \mathbf{x}) = \sum_{\mathbf{y} \neq \mathbf{x}^{(i)}} u_t(\mathbf{y}, \mathbf{x}) \, \tilde{a}_t^u(\mathbf{y}^{(i)}; \mathbf{y})$$
$$- u_t(\mathbf{x}^{(i)}, \mathbf{x}) \, \tilde{a}_t^u(\mathbf{x}; \mathbf{x}^{(i)}), \quad (20)$$

with boundary condition $\tilde{a}_1^u(\mathbf{x}'; \mathbf{x}) = g(\mathbf{x}') - g(\mathbf{x})$.

Intuitively, the "lean" adjoint associated with the state $\mathbf{x}^{(i)}$ remains unchanged if either no future flip occurs or only coordinates other than $i$ are flipped. Conversely, it changes sign whenever the $i$-th coordinate is flipped. Integrating Equation 20 yields, for any $s \in [t, 1)$,

$$\tilde{a}_t^u(\mathbf{x}^{(i)}; \mathbf{x}) = \mathbb{E}_{p_u} \Big[ (-1)^{x_i \oplus y_i} \tilde{a}_s(\mathbf{y}^{(i)}; \mathbf{y}) \Big| X_t = \mathbf{x} \Big]. \quad (21)$$

Note that this recursion does not remain accurate arbitrarily close to the terminal time, i.e., as $s \to 1$, where the adjoint interpolates between local and global improvement

directions (see Appendix A.4). Nevertheless, for local-improvement objectives, we may use the approximation

$$\tilde{a}_s^u(\mathbf{x}^{(i)}; \mathbf{x}) \approx \tilde{a}_1(\mathbf{x}^{(i)}; \mathbf{x}) \qquad \text{as } s \to 1. \quad (22)$$

Based on the above results, a single sampled trajectory $\mathbf{X}$ can be used to estimate the intermediate "lean" adjoint as

$$\tilde{a}_t^u(X_t^{(i)}; X_t) \approx (-1)^{X_{t,i} \oplus X_{1,i}} \Big( g(X_1^{(i)}) - g(X_1) \Big). \quad (23)$$

In particular, define the flipping-probability vector and flip-gradient vector as

$$\mathbf{u}_t(\mathbf{x}) = \Big( \cdots, u_t(\mathbf{x}^{(i)}, \mathbf{x}), \cdots \Big)^\top, \quad (24)$$

$$\nabla^{\text{flip}} g(\mathbf{x}) = \Big( \cdots, g(\mathbf{x}^{(i)}) - g(\mathbf{x}), \cdots \Big)^\top. \quad (25)$$

Replacing the cost-to-go difference in Equation 7 with the corresponding "lean" adjoint estimate yields our **Combinatorial Adjoint Matching (CAM)** objective:

$$\mathcal{L}_{\text{CAM}}(\mathbf{X}) = \int_{t=0}^{1} \Bigg\{ c_t(u; \mathbf{x})$$
$$+ \mathbf{u}_t(X_t)^\top \Big( (-1)^{X_t \oplus X_1} \circ \nabla^{\text{flip}} g(X_1) \Big) \Bigg\} dt. \quad (26)$$

## 4. Practical Adaptations

### 4.1. Time Discretization

In practice, the continuous time horizon $[0, 1]$ has to be discretized into a finite sequence of intervals $0 = t_0 < t_1 < \cdots < t_K = 1$. Let

$$\Lambda = \int_{s=t_k}^{t_{k+1}} \mathbf{u}_s(X_s) \, ds,$$

where $\Lambda_i$ denotes the expected number of flips of the $i$-th coordinate during $[t_k, t_{k+1}]$. When the interval is sufficiently small, we have $\Lambda_i \ll 1$, yielding the approximation

$$P_{t_k, t_{k+1}}(i\text{-th coordinate is flipped}) \approx \Lambda_i. \quad (27)$$

We therefore parameterize the integrated transition rates over each interval using a neural network $\mathbf{u}^\theta$:

$$\int_{s=t_k}^{t_{k+1}} \mathbf{u}_s(X_s) \, ds \approx \mathbf{u}_{t_k}^\theta(X_{t_k}). \quad (28)$$

Under this approximation, the transition distribution becomes a coordinate-wise Bernoulli, $\text{Ber}\big(\mathbf{u}_t^\theta(X_t)\big)$, whose integrated KL cost over the interval reduces to

$$-\tau \mathcal{H}\big(\mathbf{u}_t^\theta(X_t)\big) + \lambda \left\| \mathbf{u}_t^\theta(X_t) \right\|_1, \quad (29)$$

where $\mathcal{H}(\mathbf{u}_t^\theta(X_t))$ denotes the entropy of the Bernoulli.

Combining these components yields the discretized CAM:

$$\mathcal{L}_{\text{CAM}}(\mathbf{X}) \approx \sum_t \left\{ -\tau\mathcal{H}\big(\mathbf{u}_t^\theta(X_t)\big) + \lambda\|\mathbf{u}_t^\theta(X_t)\|_1 \right.$$
$$\left. + \mathbf{u}_t^\theta(X_t)^\top\Big((-1)^{X_t \oplus X_1} \circ \nabla^{\text{flip}}g(X_1)\Big) \right\}. \quad (30)$$

Empirically, we find that the primary performance gains of CAM originate from the final adjoint term in Equation (30), which encodes the trajectory-level optimization signal. In contrast, the entropy regularization and transition-rate penalty play a comparatively minor role. For instance, we set $\lambda = 0$ throughout our experiments and observe little performance degradation when further removing the entropy term. This observation suggests that the effectiveness of CAM is largely attributable to the discrete adjoint dynamics rather than the specific regularization induced by the stochastic control formulation. Nevertheless, we find that strong neural architectures remain important for fully realizing the benefits of CAM. Accordingly, all experiments build upon the architectures introduced in prior diffusion-based CO solvers (Sanokowski et al., 2024; 2025). Additional stabilization techniques used for weaker architectures are provided in Appendix C.

## 4.2. Computation of Flip-Gradient

The computation of the flip-gradient $\nabla^{\text{flip}}g(X_1)$ is the primary computational bottleneck in Equation 30, where iteratively evaluating $g(X_1^{(i)})$ is prohibitively expensive for large-scale problems. Below, we discuss efficient strategies for computing or approximating the flip-gradient in two representative settings, using Maximum Independent Set (MIS) and Traveling Salesman Problem (TSP) as examples.

**Quadratic Unconstrained Binary Optimization (QUBO).** Many CO problems admit relaxed formulations of the form $g(\mathbf{x}) = a(\mathbf{x}) + \beta b(\mathbf{x})$, which can be expressed in the QUBO form. This representation is particularly suitable for problems with local structural constraints, such as MIS, and constitutes the primary setting considered by most existing unsupervised diffusion-based CO solvers.

For QUBO objectives of the form

$$g(\mathbf{x}) = \mathbf{x}^\top \mathbf{Q} \mathbf{x}, \quad (31)$$

the flip-gradient admits a closed-form expression:

$$\nabla^{\text{flip}}g(\mathbf{x}) = (1 - 2\mathbf{x}) \circ (\mathbf{Q}\mathbf{x} + \mathbf{Q}^\top\mathbf{x}). \quad (32)$$

This formulation enables efficient and exact computation of the flip-gradient without explicitly enumerating all single-coordinate perturbations.

**General Combinatorial Optimization.** Training unsupervised diffusion models for general combinatorial optimization problems beyond QUBO remains largely under-explored, primarily because tractable unsupervised targets are unavailable for problems with global constraints, such as TSP. To address this challenge, we introduce a surrogate strategy for constructing the flip-gradient.

Starting from the sampled terminal solution $X_1$, we first obtain an improved local solution $X_1^+$ using a problem-specific local search heuristic. For TSP, we employ greedy decoding followed by 2-OPT refinement. We then define the surrogate objective

$$g(\mathbf{x}) = \|\mathbf{x} - X_1^+\|_1. \quad (33)$$

Under this construction, the corresponding flip-gradient becomes particularly simple: flipping a coordinate receives a positive label if it moves the current solution closer to $X_1^+$, and a negative label otherwise. Consequently, instead of regressing the magnitude of the flip-gradient, we reformulate the training objective as a binary classification problem.

Specifically, we replace the original regression-style objective with a binary cross-entropy loss that predicts whether each coordinate of $X_t$ should be flipped to match the corresponding value in $X_1^+$. This modification better aligns the learning objective with the discrete and coordinate-wise structure of the surrogate labels, while also improving training stability in practice.

## 5. Experiments

### 5.1. Experimental Setup

**CO Problems and Benchmarks.** We evaluate CAM primarily on MIS and TSP, corresponding to the two mechanisms for obtaining the flip-gradient described above.

For MIS, we evaluate on both *Revised Model B (RB)* graphs (Xu & Li, 2000) and *Erdős–Rényi (ER)* graphs across two different scales. Specifically, RB graphs are generated at small (200–300 nodes) and large (800–1,200 nodes) scales, while ER graphs include small instances with 700–800 nodes and large instances with 9,000–11,000 nodes. The large ER graphs are used as a *transfer-testing benchmark* for models trained on the small ER graphs. Throughout the paper, we use the suffix "-[m–M]" to denote the corresponding range of node counts.

For TSP, city coordinates are sampled uniformly from the unit square (Kool et al., 2019). We consider problem instances with $M = 500$ and $M = 1000$ cities, denoted as "TSP-$M$", where $M$ specifies the problem size.

**Baselines.** We primarily compare CAM against diffusion-based CO solvers that can be viewed as discrete counterparts

*Table 1.* Comparative results on Maximum Independent Set (MIS). The best results are **bolded** and the second-best ones are underlined, excluding OR solvers. The gap is computed against the result of KaMIS (Großmann et al., 2023) and CAM (on ER-[9000–11000]).

| MIS | | RB-[200–300] | | | RB-[800–1200] | | |
|---|---|---|---|---|---|---|---|
| TYPE | METHOD | SIZE ↑ | GAP ↓ | TIME ↓ | SIZE ↑ | GAP ↓ | TIME ↓ |
| OR | Gurobi | 19.98 | 0.60% | 47.57m | 40.90 | 5.21% | 2.17h |
| | KaMIS | 20.10 | 0.00% | 1.40h | 43.15 | 0.00% | 2.05h |
| SL | INTEL | 18.47 | 8.11% | 13.07m | 34.47 | 20.12% | 20.28m |
| | DGL | 17.36 | 13.93% | 12.78m | 34.50 | 20.05% | 23.90m |
| | DIFUSCO | 18.74 | 6.77% | 5.33m | 37.32 | 13.51% | 8.46m |
| UL | DiffUCO | 19.88 | 1.09% | 4.97m | 40.52 | 6.10% | 6.61m |
| | SDDS | 19.75 | 1.74% | 4.97m | 39.76 | 7.86% | 6.61m |
| | RLNN | 19.65 | 2.24% | 4.97m | 39.96 | 7.39% | 6.61m |
| | CAM (Ours) | **19.91** | **0.95%** | 4.97m | **41.25** | **4.40%** | 6.61m |

| MIS | | ER-[700–800] | | | ER-[9000–11000] | | |
|---|---|---|---|---|---|---|---|
| TYPE | METHOD | SIZE ↑ | GAP ↓ | TIME ↓ | SIZE ↑ | GAP ↓ | TIME ↓ |
| OR | Gurobi | 41.38 | 7.78% | 50.00m | — | — | — |
| | KaMIS | 44.87 | 0.80% | 52.13m | 381.31 | 0.00% | 7.60h |
| SL | INTEL | 34.86 | 22.31% | 6.06m | 284.63 | 25.95% | 5.02m |
| | DGL | 37.26 | 16.96% | 22.71m | — | — | — |
| | DIFUSCO | 41.85 | 6.73% | 1.34m | 347.00 | 9.72% | 5.89m |
| UL | DIMES | 42.06 | 6.26% | 12.01m | 332.80 | 13.42% | 12.72m |
| | DiffUCO | 43.98 | 1.98% | 55s | 373.31 | 2.88% | 5.53m |
| | SDDS | 43.31 | 3.48% | 55s | 350.63 | 8.78% | 5.53m |
| | RLNN | 43.14 | 3.86% | 55s | 374.38 | 2.60% | 5.53m |
| | CAM (Ours) | **44.16** | **1.58%** | 55s | **384.38** | **0.00%** | 5.53m |

*Table 2.* Comparative results on the Traveling Salesman Problem (TSP). The best results are **bolded** and the second-best ones are underlined, excluding OR solvers. The gap is computed against the result of LKH-3 (Helsgaun, 2017).

| TSP | | TSP-500 | | | TSP-1000 | | |
|---|---|---|---|---|---|---|---|
| TYPE | METHOD | LENGTH ↓ | GAP ↓ | TIME ↓ | LENGTH ↓ | GAP ↓ | TIME ↓ |
| OR | Concorde | 16.55 | 0.00% | 37.66m | 23.12 | 0.00% | 6.65h |
| | LKH-3 | 16.55 | 0.00% | 46.88m | 23.12 | 0.00% | 2.57h |
| SL | GCN | 30.37 | 83.55% | 38.02m | 51.26 | 121.73% | 51.67m |
| | DIFUSCO | 16.85 | 1.81% | 11.95m | 23.85 | 3.16% | 47.01m |
| | FMIP | **16.83** | **1.69%** | 11.95m | 23.81 | 2.98% | 47.01m |
| UL | AM | 19.53 | 18.03% | 21.99m | 29.90 | 29.23% | 1.64m |
| | POMO | 19.24 | 16.25% | 12.80h | 49.56 | 114.36% | 63.45h |
| | DIMES | 17.80 | 7.55% | 2.11h | 24.89 | 7.70% | 4.53h |
| | SDDS | 16.98 | 2.60% | 11.95m | 23.90 | 3.37% | 47.01m |
| | CAM (Ours) | 16.91 | 2.18% | 11.95m | **23.70** | **2.51%** | 47.01m |

of adjoint matching in continuous domains, including:

- **DIFUSCO** (Sun & Yang, 2023): standard diffusion models (Ho et al., 2020; Song et al., 2020);

- **FMIP** (Li et al., 2025): Flow Matching (Lipman et al., 2023; Liu et al., 2023);

- **DiffUCO** (Sanokowski et al., 2024): maximum-entropy reinforcement learning (Ziebart et al., 2008);

- **SDDS** (Sanokowski et al., 2025): importance-weighted matching (Domingo-Enrich et al., 2024);

- **RLNN** (Feng & Yang, 2025): Langevin dynamics (Welling & Teh, 2011; Song et al., 2021).

All diffusion-based baselines are re-implemented and trained from scratch under a standardized setup.

In addition, we include a broad range of baselines spanning operations research (OR) solvers, supervised learning (SL) methods, and unsupervised learning (UL) neural solvers. Additional details are provided in Appendix B.1.

**Post-processing.** We observe that post-processing can lead to substantial differences in final performance, even when applied to the same underlying model. To ensure a fair comparison, we standardize the post-processing procedures across all diffusion-based methods, since they share the same output format.

For TSP, we apply greedy decoding followed by 2-OPT local search. For MIS, only greedy decoding is used. Importantly, **all post-processing and evaluation are strictly restricted to the terminal state** $X_1$. For each instance, we perform multiple independent sampling runs and report the best result. All diffusion-based methods are allocated identical inference budgets in terms of both the number of inference steps and the number of independent runs.

**Metrics.** We use the original problem-specific objective as the primary evaluation metric, namely the tour length for TSP and the independent set size for MIS. For a more direct comparison, we also report the optimality gap with respect to the best achieved result: $\mathrm{gap}(c, c^*) = \frac{|c - c^*|}{|c^*|}$, where $c$ denotes the objective value achieved by a method and $c^*$ denotes the best objective value. Finally, we measure the total runtime of each method on the test set by *sequentially* processing all test instances.

### 5.2. Main Results

We summarize the main results in Tables 1 and 2.

On MIS, unsupervised diffusion-based solvers generally outperform supervised approaches. A possible explanation is that the MIS objective provides a structural local improvement signal, whose gradient is linear in $\mathbf{x}$. As a result, learning directly from the objective may be easier than regressing optimal solutions through supervision. Among all methods, CAM consistently achieves the strongest performance, often with a substantial margin over the second-best baseline, particularly on RB-[800–1200] and ER-[9000–11000]. On ER-[9000–11000], CAM even surpasses the traditional OR solver KaMIS (Großmann et al., 2023), achieving state-of-the-art (SOTA) performance.

More importantly, DiffUCO, SDDS, and RLNN require dense feedback at *every* intermediate diffusion step during training, whereas CAM operates with significantly weaker feedback. For example, CAM requires only a single MIS gradient evaluation at the terminal state to train a 50-step diffusion process, while RLNN requires the gradient feedback at all 50 intermediate states.

TSP exhibits a markedly different trend from MIS. In this setting, supervised diffusion models perform substantially better, likely because the global tour constraint makes the objective considerably harder to optimize effectively without explicit supervision. Nevertheless, despite being fully unsupervised, CAM remains highly competitive, ranking as the third-best on TSP-500 and the best on TSP-1000.

This result is particularly notable because CAM learns solely from suboptimal solutions sampled from its own policy. We additionally observe that CAM demonstrates relatively stronger performance on large-scale instances (RB-[800–1200], ER-[9000–11000], and TSP-1000) compared to the baselines, highlighting the scalability of CAM

### 5.3. Additional Evaluation

Beyond the primary benchmarks on MIS and TSP, we further evaluate CAM on additional CO problems, including Maximum Cut (Max Cut) and Capacitated Vehicle Routing Problem (CVRP), to verify whether the same empirical patterns persist across different problem families.

*Table 3.* Comparative results on Max Cut. The best results are **bolded** and the second-best ones are underlined, excluding the OR solver. The gap is computed against the result of CAM.

| | **Max Cut** | BA-[800–1200] | | |
|---|---|---|---|---|
| TYPE | METHOD | SIZE ↑ | GAP ↓ | TIME ↓ |
| OR | MQLib | 2961.38 | 0.10% | 8.33h |
| SL | DIFUSCO | 2946.23 | 0.61% | 6.08m |
| | FMIP | 2939.58 | 0.83% | 5.49m |
| UL | DiffUCO | 2961.07 | 0.11% | 5.49m |
| | SDDS | 2963.95 | 0.01% | 5.49m |
| | RLNN | 2956.81 | 0.25% | 5.49m |
| | CAM (Ours) | **2964.20** | **0.00%** | 5.49m |

**Maximum Cut.** We evaluate CAM on Max Cut over Barabási–Albert (BA) graphs (Barabási & Albert, 1999) with 800 to 1,200 nodes, as shown in Table 3. Similar to MIS, Max Cut can also be formulated as a QUBO problem, and we observe a highly consistent trend across the two tasks. In particular, unsupervised learning approaches generally achieve stronger performance than supervised diffusion-based methods on these large-scale graph optimization problems. Among all learning-based approaches, CAM achieves the best overall performance, outperforming both supervised and previous unsupervised baselines. Particularly, CAM even outperforms the OR solver MQLib (Dunning et al., 2018) with less solving time, demonstrating its effectiveness as a SOTA solver.

**Capacitated Vehicle Routing Problem.** We further evaluate CAM on the CVRP, a generalized variant of TSP with additional vehicle-capacity constraints, whose combinatorial objective is substantially more challenging to optimize. Table 4 reports the results on CVRP-100 (Ma et al., 2025b), where customer locations are uniformly sampled in a unit square. For post-processing, we decode the predicted edge scores into feasible routes using a score-guided Clarke–

*Table 4.* Comparative results on the Capacitated Vehicle Routing Problem (CVRP). The best results are **bolded** and the second-best ones are underlined, excluding OR solvers. The gap is computed against the result of the OR solver HGS (Vidal et al., 2012).

| **CVRP** | | CVRP-100 | | |
|------|--------|---------|--------|--------|
| TYPE | METHOD | LENGTH ↓ | GAP ↓ | TIME ↓ |
| OR | HGS | 15.56 | 0.00% | 55.64h |
| SL | DIFUSCO | **15.97** | **2.63%** | 33.75m |
| | FMIP | 16.04 | 3.08% | 33.70m |
| UL | SDDS | 16.16 | 3.86% | 33.70m |
| | CAM (Ours) | 15.98 | 2.70% | 33.70m |

Wright savings heuristic (Clarke & Wright, 1964), followed by a classic local search algorithm (Ma et al., 2025a). Similar to the observations on TSP, supervised approaches generally achieve stronger performance than unsupervised methods on routing problems with highly structured sequential objectives. Nevertheless, CAM significantly improves over the unsupervised baseline SDDS, outperforms the supervised method FMIP, and achieves performance close to that of the strongest supervised baseline DIFUSCO.

Overall, the empirical trends on both Max Cut and CVRP remain highly consistent with our findings on MIS and TSP.

### 5.4. Ablation Studies

We conduct several ablation studies to analyze the effects of inference-time sampling budgets, terminal heuristics, and adjoint propagation in CAM.

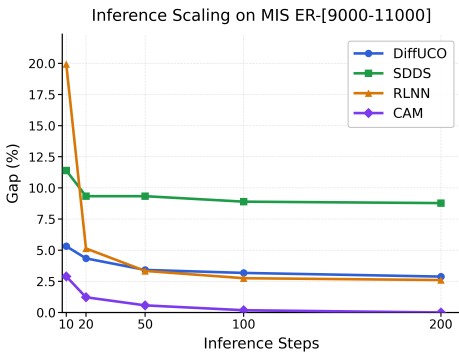

*Figure 1.* Comparison between CAM and baselines under different inference-time sampling budgets.

**Performance Across Different Sampling Steps.** Figure 1 compares CAM with several baselines under different inference-time sampling budgets on ER-[9000–11000]. CAM consistently achieves the best performance across all sampling budgets and exhibits a substantially stronger inference-time scaling effect.

In particular, even with only 10 sampling steps, CAM already achieves performance comparable to that of other

strong baselines using 200 steps. As the number of inference steps increases, CAM continues to improve and eventually reaches a zero optimality gap. These results suggest that CAM not only delivers strong performance under limited inference budgets, but also scales effectively with additional inference-time computation.

*Table 5.* Ablation on different CAM training targets on TSP-500.

| **TSP** | | TSP-500 | |
|------|------------|-----------------------|----------|
| CAM | No adjoint | Local search w/ greedy | Standard |
| GAP ↓ | 2.83% | 2.36% | **2.18%** |

**Effect of Different Training Targets.** We further investigate the role of adjoint propagation and terminal heuristic quality using several training variants on TSP-500.

Table 5 compares three variants:

- *No adjoint*, which directly imitates 2-OPT improvements without adjoint propagation;

- *Local search w/ greedy*, which removes the terminal 2-OPT post-processing during training, with only the greedy decoding for $X_1^+$; and

- The *standard* CAM objective.

Removing adjoint propagation causes a substantial degradation in performance, increasing the optimality gap from 2.18% to 2.83%. In contrast, removing the strong 2-OPT terminal heuristic only slightly degrades the final performance to 2.36%. These results indicate that the primary performance gain of CAM does not come from directly imitating powerful heuristics, but rather from propagating optimization signals backward through the trajectory via adjoint dynamics.

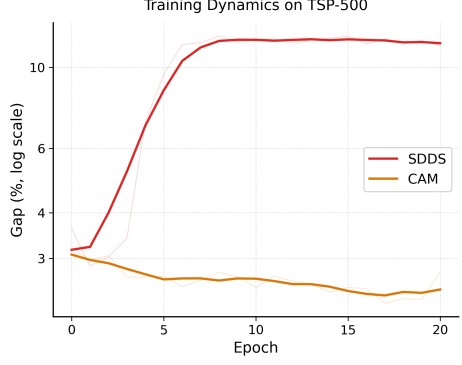

*Figure 2.* Training dynamics comparison between CAM and the unsupervised baseline SDDS.

**Comparison with SDDS.** Figure 2 compares the training dynamics between CAM and the unsupervised baseline

SDDS. While SDDS relies purely on self-sampled trajectories for optimization, CAM incorporates gradient-based trajectory guidance through adjoint propagation.

As training progresses in the high-dimensional discrete solution space, SDDS exhibits highly unstable optimization behavior and rapidly diverges. In contrast, CAM maintains stable convergence throughout training and continuously improves the solution quality. These observations suggest that trajectory-level gradient propagation provides a significantly more stable optimization signal than purely sample-based training objectives.

## 6. Related Work

### 6.1. Adjoint Method

Adjoint methods (Pontryagin et al., 1962) are a classical tool for optimizing objectives defined by Ordinary and Stochastic Differential Equations. Building on this, Adjoint Matching (AM) (Domingo-Enrich et al., 2025) frames reward-driven adaptation of continuous diffusion/flow models as a stochastic control problem, and derives a practical training objective that uses adjoint-based gradient information to update the control along the generative trajectory. A concurrent work, Discrete Adjoint Matching (DAM) (So et al., 2026), addresses a related setting but follows a fundamentally different approach. DAM relies on importance sampling to estimate the optimal control and does not propagate gradients through the trajectory, whereas CAM explicitly exploits adjoint-based gradients to capture the path-gradient structure in the combinatorial space.

### 6.2. Diffusion Models for Combinatorial Optimization

Diffusion models and their variants, originally developed for continuous data (Sohl-Dickstein et al., 2015; Ho et al., 2020; Lipman et al., 2023), have recently become effective tools for combinatorial optimization (CO). By generating solutions through iterative denoising, diffusion-based solvers can better represent multi-modal solution distributions and navigate the highly nonconvex CO landscape than one-shot generators. DIFUSCO (Sun & Yang, 2023) adapts continuous diffusion to discrete CO and substantially improves both efficiency and solution quality over prior end-to-end neural solvers. A line of work, including T2T (Li et al., 2023) and Fast T2T (Li et al., 2024), has explored improvements. More recently, FMIP (Li et al., 2025) applies flow matching (Lipman et al., 2023; Liu et al., 2023) to mixed-integer linear programs, and GenSCO (Li et al., 2026) treats diffusion sampling as a search operator by alternating solution disruption with diffusion resampling to enable efficient test-time scaling. These advances are largely orthogonal to our framework and can be incorporated independently.

A common limitation of many diffusion-based CO solvers is their reliance on supervised training with large sets of near-optimal solutions. Recent efforts explore unsupervised training, including DiffUCO (Sanokowski et al., 2024) (entropy-regularized RL) and SDDS (Sanokowski et al., 2025) (importance-weighted matching), but they typically require heavy sampling to control variance and estimate action quality. RLNN (Feng & Yang, 2025) casts diffusion as regularized Langevin dynamics and optimizes a local objective to improve scalability, but this limits the ability to capture long-term dependencies and may lead to suboptimal global solutions. In contrast, our adjoint matching approach backpropagates the terminal objective as a matching objective, providing a structured, low-variance signal that directly guides intermediate controls to improve final solution quality.

## 7. Conclusion

We presented Combinatorial Adjoint Matching (CAM), an adjoint-based framework for training diffusion solvers for combinatorial optimization in an unsupervised way. By formulating diffusion-based combinatorial optimization as a stochastic control problem over discrete trajectories, CAM enables trajectory-level optimization through discrete adjoint dynamics and structured path-gradient signals. Compared with prior unsupervised diffusion approaches based on reinforcement learning or importance-weighted matching, CAM provides a more informative and lower-variance optimization signal while substantially reducing the amount of environment feedback required during training.

Empirically, CAM consistently outperforms existing unsupervised diffusion baselines across a diverse set of combinatorial optimization problems, including Maximum Independent Set, Traveling Salesman Problem, Maximum Cut, and Capacitated Vehicle Routing Problem, while achieving performance competitive with strong supervised diffusion solvers and even traditional heuristic solvers. These results demonstrate that adjoint-based trajectory optimization can serve as an effective paradigm for scalable and data-efficient neural combinatorial optimization.

Nevertheless, several limitations remain. Our theoretical analysis relies on idealized assumptions that may not fully capture the complexity of highly non-convex NP-hard optimization landscapes. Moreover, the current framework still depends on surrogate discrete gradients and problem-specific local improvement operators to construct training signals, which may limit its applicability to more general combinatorial domains. Future work includes developing more general discrete path-gradient estimators, reducing reliance on hand-designed local search procedures, and extending adjoint-based optimization principles to broader classes of discrete generative models and combinatorial optimization problems.

## Impact Statement

This paper presents work whose goal is to advance the field of Machine Learning. There are many potential societal consequences of our work, none which we feel must be specifically highlighted here.

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

## A. Proofs of Theoretical Results

We first define the value function associated with the SOC problem as optimal cost-to-go:

$$V_t(\mathbf{x}) := \min_u J_t(u; \mathbf{x}). \tag{34}$$

For the reference CTMC process $p^{\text{base}}$ (Kappen, 2005), the value function admits the following exponential expectation representation (Bellman, 1954):

$$V_t(\mathbf{x}) = -\tau \log \mathbb{E}_{p^{\text{base}}} \left[ \exp\left( -\frac{g(X_1)}{\tau} \right) \bigg| X_t = \mathbf{x} \right]. \tag{35}$$

### A.1. Proof of Proposition 3.1

Let $\mathbf{x}^\star$ be the closest minimizer to $\mathbf{x}$, and denote

$$d = \|\mathbf{x} - \mathbf{x}^\star\|_1. \tag{36}$$

Then Equation 35 becomes

$$
\begin{aligned}
V_t(\mathbf{x}) &= -\tau \log \left[ \exp\left( -\frac{g^\star}{\tau} \right) \mathbb{E}_{p^{\text{base}}} \left[ \exp\left( \frac{g^\star - g(X_1)}{\tau} \right) \bigg| X_t = \mathbf{x} \right] \right] \\
&= g^\star - \tau \log \mathbb{E}_{p^{\text{base}}} \left[ \exp\left( \frac{g^\star - g(X_1)}{\tau} \right) \bigg| X_t = \mathbf{x} \right].
\end{aligned}
\tag{37}
$$

Under the homogeneous base process, each coordinate evolves independently. For a coordinate that agrees with $\mathbf{x}^\star$ at time $t$, the probability that it agrees with $\mathbf{x}^\star$ at time 1 is

$$\frac{1 + \exp(-2(1-t)\rho_\tau)}{2}. \tag{38}$$

For a coordinate that disagrees with $\mathbf{x}^\star$ at time $t$, the probability that it agrees with $\mathbf{x}^\star$ at time 1 is

$$\frac{1 - \exp(-2(1-t)\rho_\tau)}{2}. \tag{39}$$

Therefore,

$$p^{\text{base}}(X_1 = \mathbf{x}^\star \mid X_t = \mathbf{x}) = \left( \frac{1 + \exp(-2(1-t)\rho_\tau)}{2} \right)^{N-d} \left( \frac{1 - \exp(-2(1-t)\rho_\tau)}{2} \right)^d. \tag{40}$$

Since

$$\rho_\tau = \sigma(-\lambda/\tau) = \exp(-\lambda/\tau)(1 + o(1)), \tag{41}$$

we have

$$\frac{1 + \exp(-2(1-t)\rho_\tau)}{2} = 1 + o(1), \tag{42}$$

and

$$\frac{1 - \exp(-2(1-t)\rho_\tau)}{2} = (1-t)\rho_\tau(1 + o(1)). \tag{43}$$

Hence

$$p^{\text{base}}(X_1 = \mathbf{x}^\star \mid X_t = \mathbf{x}) = (1-t)^d \exp\left( -\frac{d\lambda}{\tau} \right) (1 + o(1)). \tag{44}$$

Assume $\lambda > 0$ is sufficiently small so that, in the low-temperature limit, the contribution of the closest minimizer $\mathbf{x}^\star$ dominates the exponential expectation. Then

$$
\begin{aligned}
V_t(\mathbf{x}) &= g^\star - \tau \log \left[ (1-t)^d \exp\left( -\frac{d\lambda}{\tau} \right) (1 + o(1)) \right] \\
&= g^\star + d\lambda - \tau d \log(1-t) + o(\tau).
\end{aligned}
\tag{45}
$$

Now consider the neighboring state $\mathbf{x}^{(i)}$, obtained by flipping the $i$-th coordinate of $\mathbf{x}$. There are two cases.

**Case 1:** $x_i \neq x_i^\star$. Flipping coordinate $i$ moves $\mathbf{x}$ closer to $\mathbf{x}^\star$, so

$$\|\mathbf{x}^{(i)}, \mathbf{x}^\star\| = d - 1. \tag{46}$$

Therefore,

$$V_t(\mathbf{x}^{(i)}) = g^\star + (d-1)\lambda - \tau(d-1)\log(1-t) + o(\tau), \tag{47}$$

and hence

$$V_t(\mathbf{x}^{(i)}) - V_t(\mathbf{x}) = -\lambda + \tau\log(1-t) + o(\tau). \tag{48}$$

Using

$$u_t^\star(\mathbf{x}^{(i)}, \mathbf{x}) = u_t^{\text{base}}(\mathbf{x}^{(i)}, \mathbf{x}) \exp\left(-\frac{V_t(\mathbf{x}^{(i)}) - V_t(\mathbf{x})}{\tau}\right), \tag{49}$$

and

$$u_t^{\text{base}}(\mathbf{x}^{(i)}, \mathbf{x}) = \rho_\tau = \exp(-\lambda/\tau)(1 + o(1)), \tag{50}$$

we obtain

$$\begin{aligned}
u_t^\star(\mathbf{x}^{(i)}, \mathbf{x}) &= \exp\left(-\frac{\lambda}{\tau}\right)\exp\left(\frac{\lambda}{\tau} - \log(1-t) + o(1)\right) \\
&= \frac{1}{1-t}(1 + o(1)).
\end{aligned} \tag{51}$$

Thus

$$u_t^\star(\mathbf{x}^{(i)}, \mathbf{x}) \to \frac{1}{1-t}. \tag{52}$$

**Case 2:** $x_i = x_i^\star$. Flipping coordinate $i$ moves $\mathbf{x}$ farther from $\mathbf{x}^\star$, so

$$\|\mathbf{x}^{(i)} - \mathbf{x}^\star\| = d + 1. \tag{53}$$

Therefore,

$$V_t(\mathbf{x}^{(i)}) = g^\star + (d+1)\lambda - \tau(d+1)\log(1-t) + o(\tau), \tag{54}$$

and hence

$$V_t(\mathbf{x}^{(i)}) - V_t(\mathbf{x}) = \lambda - \tau\log(1-t) + o(\tau). \tag{55}$$

Again using the optimal-rate formula,

$$\begin{aligned}
u_t^\star(\mathbf{x}^{(i)}, \mathbf{x}) &= \exp\left(-\frac{\lambda}{\tau}\right)\exp\left(-\frac{\lambda}{\tau} + \log(1-t) + o(1)\right) \\
&= (1-t)\exp\left(-\frac{2\lambda}{\tau}\right)(1 + o(1)).
\end{aligned} \tag{56}$$

Thus

$$u_t^\star(\mathbf{x}^{(i)}, \mathbf{x}) \to 0. \tag{57}$$

Combining the two cases gives

$$u_t^\star(\mathbf{x}^{(i)}, \mathbf{x}) = \begin{cases} \dfrac{1}{1-t}, & x_i \neq x_i^\star, \\ 0, & x_i = x_i^\star, \end{cases} \tag{58}$$

as $\tau \to 0$, which proves the proposition. $\qquad\square$

## A.2. Proof of Proposition 3.2

Recall the expected future cost under a fixed policy $u$:

$$J_t(u; \mathbf{x}) = \mathbb{E}_{p^u}\left[g(X_1) + \int_t^1 c_s(u; X_s)\, ds \,\Big|\, X_t = \mathbf{x}\right]. \tag{59}$$

Then $J_t(u; \mathbf{x})$ satisfies the backward Kolmogorov equation

$$-\partial_t J_t(u; \mathbf{x}) = c_t(u; \mathbf{x}) + \sum_{\mathbf{y} \neq \mathbf{x}} u_t(\mathbf{y}, \mathbf{x})\big(J_t(u; \mathbf{y}) - J_t(u; \mathbf{x})\big). \tag{60}$$

By definition, the discrete adjoint is

$$a_t^u(\mathbf{x}'; \mathbf{x}) = J_t(u; \mathbf{x}') - J_t(u; \mathbf{x}). \tag{61}$$

Taking the time derivative gives

$$-\partial_t a_t^u(\mathbf{x}'; \mathbf{x}) = -\partial_t J_t(u; \mathbf{x}') + \partial_t J_t(u; \mathbf{x}). \tag{62}$$

Substituting Equation 60 for both states yields

$$-\partial_t a_t^u(\mathbf{x}'; \mathbf{x}) = \sum_{\mathbf{y} \neq \mathbf{x}'} u_t(\mathbf{y}, \mathbf{x}')\big(J_t(u; \mathbf{y}) - J_t(u; \mathbf{x}')\big) - \sum_{\mathbf{y} \neq \mathbf{x}} u_t(\mathbf{y}, \mathbf{x})\big(J_t(u; \mathbf{y}) - J_t(u; \mathbf{x})\big) + c_t(u; \mathbf{x}') - c_t(u; \mathbf{x}). \tag{63}$$

Using again the definition of the discrete adjoint,

$$-\partial_t a_t^u(\mathbf{x}'; \mathbf{x}) = \sum_{\mathbf{y} \neq \mathbf{x}'} u_t(\mathbf{y}, \mathbf{x}') a_t^u(\mathbf{y}; \mathbf{x}') - \sum_{\mathbf{y} \neq \mathbf{x}} u_t(\mathbf{y}, \mathbf{x}) a_t^u(\mathbf{y}; \mathbf{x}) + c_t(u; \mathbf{x}') - c_t(u; \mathbf{x}). \tag{64}$$

This proves Proposition 3.2. $\square$

## A.3. Proof of Proposition 3.3

Under Proposition 3.1, the optimal value satisfies $J_t(u^\star; \mathbf{x}) = V_t(\mathbf{x})$. Therefore,

$$a_t^{u^\star}(\mathbf{x}; \mathbf{x}^{(i)}) = J_t(u^\star; \mathbf{x}) - J_t(u^\star; \mathbf{x}^{(i)}) = V_t(\mathbf{x}) - V_t(\mathbf{x}^{(i)}). \tag{65}$$

Let $\mathbf{x}^\star$ be the closest minimizer to $\mathbf{x}$. There are two cases.

**Case 1: $x_i \neq x_i^\star$.** Flipping coordinate $i$ moves $\mathbf{x}$ closer to $\mathbf{x}^\star$. From Proposition 3.1,

$$u_t^\star(\mathbf{x}^{(i)}, \mathbf{x}) = \frac{1}{1-t} + o(1), \qquad u_t^\star(\mathbf{x}, \mathbf{x}^{(i)}) = 0 + o(1). \tag{66}$$

Moreover, from the low-temperature expansion of $V_t$,

$$V_t(\mathbf{x}) - V_t(\mathbf{x}^{(i)}) = \lambda - \tau \log(1-t) + o(\tau). \tag{67}$$

Thus

$$a_t^{u^\star}(\mathbf{x}; \mathbf{x}^{(i)}) = \lambda - \tau \log(1-t) + o(\tau). \tag{68}$$

The limiting running cost is

$$c_t(u^\star; \mathbf{x}) = \lambda \sum_{\mathbf{y} \neq \mathbf{x}} u_t^\star(\mathbf{y}, \mathbf{x}) + o(1). \tag{69}$$

Since $\mathbf{x}$ has one more mismatched coordinate than $\mathbf{x}^{(i)}$, we have

$$c_t(u^\star; \mathbf{x}^{(i)}) - c_t(u^\star; \mathbf{x}) = -\frac{\lambda}{1-t} + o(1). \tag{70}$$

Therefore,

$$u_t^\star(\mathbf{x}, \mathbf{x}^{(i)}) a_t^{u^\star}(\mathbf{x}; \mathbf{x}^{(i)}) + c_t(u^\star; \mathbf{x}^{(i)}) - c_t(u^\star; \mathbf{x}) = -\frac{\lambda}{1-t} + o(1), \tag{71}$$

while

$$
\begin{aligned}
&- u_t^\star(\mathbf{x}^{(i)}, \mathbf{x}) a_t^{u^\star}(\mathbf{x}; \mathbf{x}^{(i)}) \\
&= -\left(\frac{1}{1-t} + o(1)\right)(\lambda - \tau \log(1-t) + o(\tau)) \\
&= -\frac{\lambda}{1-t} + o(1).
\end{aligned}
\tag{72}
$$

Thus the desired identity holds in the low-temperature limit.

**Case 2: $x_i = x_i^\star$.** Flipping coordinate $i$ moves $\mathbf{x}$ farther from $\mathbf{x}^\star$. From Proposition 3.1,

$$u_t^\star(\mathbf{x}^{(i)}, \mathbf{x}) = 0 + o(1), \qquad u_t^\star(\mathbf{x}, \mathbf{x}^{(i)}) = \frac{1}{1-t} + o(1). \tag{73}$$

Moreover, from the low-temperature expansion of $V_t$,

$$V_t(\mathbf{x}) - V_t(\mathbf{x}^{(i)}) = -\lambda + \tau \log(1-t) + o(\tau). \tag{74}$$

Thus

$$a_t^{u^\star}(\mathbf{x}; \mathbf{x}^{(i)}) = -\lambda + \tau \log(1-t) + o(\tau). \tag{75}$$

Since $\mathbf{x}^{(i)}$ has one more mismatched coordinate than $\mathbf{x}$, we have

$$c_t(u^\star; \mathbf{x}^{(i)}) - c_t(u^\star; \mathbf{x}) = \frac{\lambda}{1-t} + o(1). \tag{76}$$

Therefore,

$$
\begin{aligned}
&u_t^\star(\mathbf{x}, \mathbf{x}^{(i)}) a_t^{u^\star}(\mathbf{x}; \mathbf{x}^{(i)}) + c_t(u^\star; \mathbf{x}^{(i)}) - c_t(u^\star; \mathbf{x}) \\
&= \left(\frac{1}{1-t} + o(1)\right)(-\lambda + \tau \log(1-t) + o(\tau)) + \frac{\lambda}{1-t} + o(1) \\
&= o(1),
\end{aligned}
\tag{77}
$$

while

$$-u_t^\star(\mathbf{x}^{(i)}, \mathbf{x}) a_t^{u^\star}(\mathbf{x}; \mathbf{x}^{(i)}) = o(1). \tag{78}$$

Thus the desired identity again holds in the low-temperature limit.

Combining the two cases gives

$$u_t^\star(\mathbf{x}, \mathbf{x}^{(i)}) a_t^{u^\star}(\mathbf{x}; \mathbf{x}^{(i)}) + c_t(u^\star; \mathbf{x}^{(i)}) - c_t(u^\star; \mathbf{x}) - u_t^\star(\mathbf{x}^{(i)}, \mathbf{x}) a_t^{u^\star}(\mathbf{x}; \mathbf{x}^{(i)}) + o(1), \tag{79}$$

as $\tau \to 0$. Equivalently,

$$\lim_{\tau \to 0}\left[ u_t^\star(\mathbf{x}, \mathbf{x}^{(i)}) a_t^{u^\star}(\mathbf{x}; \mathbf{x}^{(i)}) + c_t(u^\star; \mathbf{x}^{(i)}) - c_t(u^\star; \mathbf{x}) + u_t^\star(\mathbf{x}^{(i)}, \mathbf{x}) a_t^{u^\star}(\mathbf{x}; \mathbf{x}^{(i)}) \right] = 0. \tag{80}$$

This proves Proposition 3.3. $\qquad\square$

## A.4. Remark: Applicability of Proposition 3.3

We note that Proposition 3.3 describes the asymptotic behavior in the low-temperature regime with a fixed intermediate time $t < 1$. In particular, the proposition does not necessarily hold near the terminal time $t \to 1$.

Recall that

$$V_t(\mathbf{x}) = g^\star - \tau \log \mathbb{E}_{p^{\text{base}}} \left[ \exp \left( \frac{g^\star - g(X_1)}{\tau} \right) \Big| X_t = \mathbf{x} \right]. \tag{81}$$

In the proof of Proposition 3.1, we showed that the contribution of the closest minimizer $\mathbf{x}^\star$ scales as

$$(1 - t)^d \exp \left( -\frac{d\lambda}{\tau} \right) (1 + o(1)), \tag{82}$$

where $d = \|\mathbf{x} - \mathbf{x}^\star\|_1$ is the Hamming distance to the closest minimizer. For fixed $t < 1$ and sufficiently small $\tau$, this term dominates the expectation, leading to the global shortest-path objective

$$V_t(\mathbf{x}) \approx g^\star + d\lambda. \tag{83}$$

However, for any fixed $\tau > 0$, if $t \to 1$, then the factor $(1 - t)^d$ vanishes. In this regime, there exist terminal states $X_1$ such that

$$(1 - t) \ll \exp \left( \frac{g^\star - g(X_1)}{\tau} \right), \tag{84}$$

so the dominant contribution to the expectation no longer comes from the closest minimizer. Instead, the expectation becomes dominated by terminal states that are locally reachable near time 1. As a result,

$$V_t(\mathbf{x}) \approx g(\mathbf{x}), \qquad t \to 1. \tag{85}$$

Therefore, the value function interpolates between two regimes:

$$V_t(\mathbf{x}) \approx \begin{cases} g^\star + d\lambda, & (1 - t) \gg \exp \left( \frac{g^\star - g(X_1)}{\tau} \right), \\ g(\mathbf{x}), & (1 - t) \ll \exp \left( \frac{g^\star - g(X_1)}{\tau} \right). \end{cases} \tag{86}$$

The former corresponds to a global shortest-path objective toward the optimal solution, while the latter corresponds to a purely local improvement objective.

Consequently, the discrete adjoint also interpolates between these two extremes. In the global regime,

$$a_t(\mathbf{x}^{(i)}; \mathbf{x}) \approx (-1)^{x_i \oplus x_i^\star + 1} \lambda, \tag{87}$$

which depends only on whether coordinate $i$ agrees with the closest minimizer. In contrast, near the terminal time,

$$a_t(\mathbf{x}^{(i)}; \mathbf{x}) \approx g(\mathbf{x}^{(i)}) - g(\mathbf{x}), \tag{88}$$

which reduces to the local one-step improvement objective.

The recursion in Proposition 3.3 therefore captures the global low-temperature structure away from the terminal boundary, whereas near $t = 1$, the adjoint naturally degenerates to the local terminal objective used in combinatorial local search. In our approximation, we **implicitly assume the consistency between the local improvement direction and the global improvement direction** in Equation 22, so additional design should be introduced to prevent the local optima (e.g., the regularization) if the assumption is severely violated.

## B. Additional Experimental Details

### B.1. Summary of Baseline Choices

We summarize all baselines used in our experiments below.

**Maximum Independent Set (MIS).** For MIS, we consider three categories of baselines.

- **OR solvers**: Gurobi (Gurobi Optimization, LLC, 2023) and KaMIS (Großmann et al., 2023).

- **Supervised methods**: INTEL (Li et al., 2018), DGL (Böther et al., 2022), and DIFUSCO.

- **Unsupervised methods**: DIMES, DiffUCO (Sanokowski et al., 2024), SDDS (Sanokowski et al., 2025), and RLNN (Feng & Yang, 2025).

**Traveling Salesman Problem (TSP).** For TSP, we adopt the following baselines.

- **OR solvers**: Concorde (Applegate et al., 2006) and LKH-3.

- **Supervised methods**: GCN (Joshi et al., 2019), DIFUSCO (Sun et al., 2022), and FMIP (Li et al., 2025).

- **Unsupervised / RL-based methods**: AM (Kool et al., 2019), POMP (Kwon et al., 2020), DIMES (Qiu et al., 2022), and SDDS (Sanokowski et al., 2025).

For SDDS, we use the forward-KL variant based on importance sampling. Following Sanokowski et al. (2024), the denoising distribution is selected from either `Bernoulli` or `Annealing`, and we report the variant that achieves the best performance on each dataset (mostly `Annealing`).

For non-diffusion baselines, we directly report the results from the original papers (Sun & Yang, 2023; Feng & Yang, 2025). All diffusion-based baselines are re-implemented and trained under a unified experimental setup.

Several variants and follow-up models of these diffusion baselines are discussed in Section 6. These approaches typically introduce additional optimizations through search heuristics or hybrid supervised–unsupervised objectives. Since our primary goal is to evaluate the effectiveness of the proposed diffusion paradigms, particularly as unsupervised frameworks, we do not include direct comparisons with these variants. Such techniques are largely orthogonal and could, in principle, be integrated into our method as well.

### B.2. Implementation Details

We summarize the implementation details used for training and evaluating all diffusion-based models.

**MIS.** Following Sanokowski et al. (2024); Qiu et al. (2022), we generate 4,000 training instances and 1,000 testing instances for RB graphs at both scales. For ER graphs, we use 16,284 training instances, together with 128 test instances for ER-[700–800] and 16 test instances for ER-[9000–11000]. KaMIS (Großmann et al., 2023) is used to generate supervision for supervised baselines.

All diffusion-based methods employ an encode–process–decode graph neural network architecture adapted from prior diffusion-based combinatorial optimization frameworks (Sanokowski et al., 2024; 2025). We omit the timestep embedding in all models, as we empirically observe that it consistently degrades performance.

The encoder maps the scalar node state into a hidden representation:

$$\mathbf{H}^0 = \text{Encoder}(\mathbf{X}), \tag{89}$$

where $\mathbf{X} \in \mathbb{R}^{N \times 1}$ denotes the current solution state. The encoder is implemented as a two-layer MLP with ReLU activations and LayerNorm.

For message passing, we employ a linear neighborhood aggregation operator with normalized adjacency. Let

$$\hat{\mathbf{A}} = \mathbf{A} + \mathbf{I}, \tag{90}$$

and

$$\tilde{\mathbf{A}} = \mathbf{D}^{-1/2} \hat{\mathbf{A}}, \tag{91}$$

where $\mathbf{D}$ denotes the degree matrix associated with $\hat{\mathbf{A}}$.

At layer $l$, neighbor messages are computed as

$$\mathbf{M}^l = \tilde{\mathbf{A}}\mathbf{W}^l_{\mathrm{msg}}\mathbf{H}^l, \tag{92}$$

and concatenated with the current node representation:

$$\mathbf{Z}^l = \mathrm{MLP}^l\left([\mathbf{H}^l, \mathbf{M}^l]\right). \tag{93}$$

The hidden states are then updated via

$$\mathbf{H}^{l+1} = \mathrm{LayerNorm}\left(\mathbf{W}^l_{\mathrm{node}}\mathbf{H}^l + \mathbf{Z}^l\right). \tag{94}$$

Unless otherwise specified, we use 8 message-passing layers with hidden dimension 64. For RB-[800–1200], we use 6 layers to improve computational efficiency. GraphNorm (Cai et al., 2021) is applied after neighborhood aggregation at every layer. The final node-wise logits are produced by a two-layer MLP decoder followed by a linear prediction head.

For optimization, DIFUSCO, RLNN, and CAM use AdamW (Loshchilov & Hutter, 2019). DiffUCO and SDDS follow the original implementation (Sanokowski et al., 2024) and use RAdam (Liu et al., 2020) with gradient clipping at a maximum norm of 1.0. Weight decay is fixed to $1 \times 10^{-4}$ for all methods, while learning rates are selected from the range $[10^{-4}, 10^{-3}]$.

For unsupervised diffusion models, the temperature parameter $\tau$ is linearly annealed from an initial value $\tau_0 \in \{0.1, 0.05\}$ to 0 throughout training.

During inference, all methods use 50 diffusion steps and 20 independent sampling runs. For ER-[9000–11000], we increase the sampling budget to 200 diffusion steps.

**TSP.** All TSP models follow the default configuration of DIFUSCO (Sun & Yang, 2023) unless otherwise specified.

Following Sun & Yang (2023), we generate 128,000 training instances for TSP-500 and 64,000 training instances for TSP-1000. LKH-3 (Helsgaun, 2017) is used to generate supervision for DIFUSCO and FMIP. We evaluate all methods on 128 test instances.

For the model architecture, we adopt a 12-layer Anisotropic Graph Neural Network with hidden dimension 256. Optimization is performed using AdamW with learning rate $2 \times 10^{-4}$ and weight decay $1 \times 10^{-4}$. All models are trained for 50 epochs. The batch size is set to 8 for TSP-500 and 4 for TSP-1000.

During training, all unsupervised diffusion models generate 4 trajectories using 10 diffusion steps. DIFUSCO uses 1,000 diffusion steps following the original implementation. A linear diffusion schedule is used for both DIFUSCO and FMIP.

During inference, all methods use 10 diffusion steps with 16 independent samples. We further apply 1,000 iterations of 2-OPT local search for both TSP-500 and TSP-1000. For DIFUSCO and FMIP, a cosine diffusion schedule is adopted at test time.

The experimental setups for Max Cut and CVRP largely follow those used for MIS and TSP, respectively. Detailed implementation choices are provided in our official codebase.

## C. Additional Training Techniques

This section describes several auxiliary training techniques that we explored during the development of CAM. Although the final results reported in the main paper are obtained using the architectures adopted from prior diffusion-based CO solvers, we found the following techniques useful for improving optimization stability and mitigating local-optima issues under certain training settings.

### C.1. Update Magnitude Regularization

Since CAM is ultimately driven by local improvement signals, the optimization process may occasionally become trapped in poor local optima, particularly during the early stages of training. Following Feng & Yang (2025), one way to encourage exploration is to regularize the expected update magnitude at each diffusion step.

Specifically, we replace the transition-rate penalty with the following regularization term:

$$\lambda\|\mathbf{u}_t^\theta(X_t)\|_1 \quad \Rightarrow \quad \alpha\big(\|\mathbf{u}_t^\theta(X_t)\|_1 - \Delta\big)^2, \tag{95}$$

which encourages the expected number of coordinate flips to remain close to a target value $\Delta > 0$, with $\alpha$ controlling the strength of the penalty.

Interestingly, when $\Delta = \|X_0 - \mathbf{x}^*\|_1/K$, with $\mathbf{x}^*$ the closest minimizer to $X_0$, the optimality condition remains unchanged, preserving the theoretical properties established in Section 3. In practice, this regularization encourages exploration by preventing the transition rates from collapsing prematurely during training.

### C.2. Balancing Local and Global Training Objectives

Although CAM propagates trajectory-level optimization signals, training may still suffer from insufficient intermediate feedback in some settings. To alleviate this issue, we consider a hybrid objective that interpolates between local and global supervision.

Specifically, we partition the diffusion trajectory into $L$ equal segments:

$$\big[0, \tfrac{1}{L}\big], \quad \big[\tfrac{1}{L}, \tfrac{2}{L}\big], \quad \cdots, \quad \big[\tfrac{L-1}{L}, 1\big], \tag{96}$$

and apply the CAM objective independently within each segment. Concretely, this requires evaluating the flip-gradient at the terminal state of every segment, resulting in $L$ terminal evaluations instead of a single evaluation at $t = 1$.

This modification introduces additional local optimization signals while preserving the overall trajectory-level structure of CAM, thereby providing a practical trade-off between optimization stability and feedback efficiency.

