# OpenReview forum: "Unsupervised Diffusion Solver for Combinatorial Optimization via Combinatorial Adjoint Matching"
_ICML.cc/2026/Conference — ICML 2026 regular_

### Official Review · Reviewer_Ycfr · 2026-03-07

**Soundness:** 2
**Presentation:** 2
**Significance:** 2
**Originality:** 3
**Overall Recommendation:** 4
**Confidence:** 1

**Summary:**

This paper looks at learning neural solvers for combinatorial optimisation using unsupervised learning and diffusion models. The approach is based on Adjoint Matching, which is an unsupervised diffusion framework in continuous spaces for propagating gradient information from a terminal objective backward through the generative process. The core innovation of the paper seems to be in extending the Adjoint Matching framework to discrete spaces, where standard gradients and the chain rule break.

More specifically, the paper model the diffusion over binary solutions as a CTMC with a learnable flip rate control $u_t(x)$ over 1 Hamming neighbours. They define a stochastic optimal control objective which combines a running flip cost with a terminal CO cost. To handle the non-differentiability problem, they introduce a flip gradient and define a combinatorial adjoint state. They show that this leads to a "lean" discrete adjoint that lets training use terminal flip gradient information as the learning signal for intermediate controls. Moreover, they also discuss several practical adaptations to make their approach work in practice (e.g. discretised training objective with regularisation).

Empirically, they evaluate on the Travelling Salesman problem and Maximum Independent Set problem. They find that they obtain competitive, or better, performance than supervised diffusion solvers.

**Compliance With Llm Reviewing Policy:**

Affirmed.

**Final Justification:**

The authors addressed my concerns. I maintain my positive score and recommend acceptance, noting that I'm not familiar with this area of research.

**Key Questions For Authors:**

Q1) It is not clear whether all methods are using the same number of independent samples (or sample wall clock budget). Can you please clarify if this is the case or not, and if not, please provide performance vs. number of samples (or time) curves for the different approaches.

Q2) For TSP, how is the method optimising the original objective vs just doing local search? More specifically, how is the $x'$ created during training and how sensitive are the results to this choice? Can you provide an ablation to quantify how much of the improvement actually comes from the adjoint propagation vs. just learning a better local improvement operator?

Q3) It would help if the authors could quantify the uncertainty/statistical significance of their results, especially as they perform close to other baselines in many cases.

**Limitations:**

No. The paper does not properly discuss its limitations. In particular they can mention their dependence on local search structure as well as a good heuristic surrogate target. Moreover, the paper does not discuss the compute tradeoffs of their approach (e.g. the extra compute that comes from increased terminal flip gradient evaluations during trajectory clipping).

**Strengths And Weaknesses:**

# Soundness

While I am not familiar with this area, the paper appears sound overall.

## Strengths:
- Their derivation for the discrete version of adjoint matching using their flip based gradient seems reasonable and well-motivated. The definition of the flip gradient and the CAM objective also seems reasonable.
- Empirically, it looks like the method is evaluated on two standard CO benchmarks with a broad range of baselines. The paper reports both performance metrics and runtime.

## Weaknesses:
- The paper mentions in the appendix that they exclude several SOTA baselines that add additional engineering to focus on the "diffusion paradigm". I feel that they should better scope their claim of SOTA to account for this.
- I am not too familiar with this area and whether it is standard practice or reasonable to report standard errors. But it would help if the authors could quantify the uncertainty/statistical significance of their results, especially as they perform close to other baselines in many cases.


# Presentation:

Overall the paper is very well presented, with a clear structure and narrative.

## Weaknesses:
It looks like there are some typos in the paper:
- The paper consistently uses $0, 1^n$ in e.g. Eqn. 16.
- "We We evaluate.." on line 303
- "I set the..." should be "We set the...." on line 318-319.
- "trained from the scratch" should be "trained from scratch" on line 287-288.
- Remove one "required" from line 85.
- Wrong capitalisation on the second sentence of line 654.


# Significance:

The paper tackles and interesting problem of learning unsupervised solvers for CO, where collecting a dataset of near-optimal solutions for supervised learning can be expensive. Their adjoint matching approach also tackles an important problem with previous unsupervised approaches which rely on high variance RL rewards.

## Weaknesses:
- It seems like the approach is best suited when the terminal flip gradient is actually tractable (e.g. in QUBO).
- I'm not too familiar with this line of research, but it seems as though the experimental scope is quite narrow (i.e. Travelling Salesman Problem and MIS) and toy. How well does the approach work for other important CO problems where good local  search may be less straightforward?


# Originality:

The main novelty of the paper is in extending adjoint matching to discrete combinatorial spaces with CTMC diffusion dynamics and their idea of flip gradients. They describe conditions under which a discrete "lean adjoint" becomes tractable and gives a simple training loss.

---

> ### Author Rebuttal · Authors · 2026-03-31
>
> We want to thank the reviewer for helpful feedback. **Additional results are accessible from this [link](https://drive.google.com/file/d/1ehRcd0WI1R8mYWQi22s3rB_nXGFfCreu/view?usp=drive_link)**.
>
> Here are the detailed response:
>
> ---
>
> >  it would help if the authors could quantify the uncertainty/statistical significance of their results
>
> Thank you for the suggestion. Since prior work does not report standard errors, we cannot uniformly quantify uncertainty across all methods. Instead, we conduct paired-t tests in the most relevant cases where CAM is within 1% of the second-best baseline (RB-[200–300], TSP-1000):
>
> | Dataset | Comparison | Metric | Value |
> |----------------|------------------|----------|---------|
> | RB-[200-300] | CAM (1.74%) vs. RLNN (1.94%) | p-value | 0.16 |
> | TSP-1000 | CAM (2.51%) vs. FM (2.98%) | p-value | 0.0007 |
>
> On RB-[200–300], the performance of CAM is close to RLNN, and the p-value (0.16) also justifies this. This is likely due to the small scale of the dataset, where performance is near saturation. But considering the reduced reliance on intermediate signals and stronger results on RB-[800–1200], we believe CAM is overall more effective than RLNN.
>
> On TSP-1000, the improvement is statistically significant (p = 0.0007).
>
> Overall, CAM matches strong baselines in near-saturated regimes and shows clearer gains on larger instances.
>
> ---
>
> > I'm not too familiar with this line of research, but it seems as though the experimental scope is quite narrow (i.e. Travelling Salesman Problem and MIS) and toy.
>
> Thank you for your question. Our evaluation **is consistent with the evaluation scope from the prior works** [1,2].
> But following your suggestion, we further extend our evaluation to the capacitated vehicle routing problem (CVRP) and  the MaxCut problem. Please see Section A at the above link.
>
> To summarize the results:
> - On CVRP, our proposed CAM has achieved comparable performance to the existing supervised diffusion models, and outperforms supervised flow matching and the unsupervised SDDS baselines.
> - On MaxCut, CAM even outperforms the human-designed heuristic solver.
>
> ---
>
> > How well does the approach work for other important CO problems where good local search may be less straightforward?
>
> Thank you for the question. We want to clarify that the **heuristic local search method is needed in all the existing neural solvers**, in order to transform the output continuous logits into a feasible solution with discrete variables.
>
> For arbitrary CO problems, some problem-invariant local search heuristics like Monte Carlo Tree Search (MCTS) could be used, which is even stronger than the 2-OPT heuristic we use [1,2] . So such a reliance is not a concern for CAM.
>
> ---
>
> > For TSP, ... how is the x’ created during training and how sensitive are the results to this choice? Can you provide an ablation to quantify how much of the improvement actually comes from the adjoint propagation vs. just learning a better local improvement operator?
>
> In our implementation, $x’$ is obtained by applying greedy decoding followed by 2-opt refinement to the sampled terminal solution $x_1$. This is necessary because TSP does not admit a tractable objective, requiring a practical surrogate for terminal improvement.
>
> To test the sensitivity and distinguish whether CAM is optimizing the objective or simply learning a better local search heuristic, we perform an ablation on TSP-500 with two variants
> - *No adjoint*, which directly imitates 2-opt improvements without adjoint propagation, and
> - *Local search w/ greedy*, which keeps CAM but replaces 2-opt with a weaker greedy heuristic.
>
> We find **removing adjoint propagation significantly degrades performance**, worse than using a weaker heuristic in CAM (Table 4, Figure 3 in the above link, partially in the Table below).
>
> | Setting | No adjoint | Local search w/ greedy | Standard CAM |
> |--------|------------|------------------------|----------|
> | GAP ↓ | 2.83% | 2.36% | **2.18%** |
>
> These results demonstrate that **CAM’s advantage comes from adjoint propagation**, rather than local search imitation.
>
> ---
>
> > It is not clear whether all methods are using the same number of independent samples (or sample wall clock budget).
>
> Thank you for your suggestion. All our diffusion models on TSP use the same model architecture and the same number of sampling budgets.
>
> On MIS, our reproduction respects the configuration choice of each method, so it is not standardized. Following your suggestion, we fully standardize the network architecture and the number of the sampling steps/samples. The visualized plot is available at Figure 2 from the above link.
>
> To summarize, CAM achieves the best performance and shows better scaling with more diffusion steps.
>
> [1] DIFUSCO: Graph-based Diffusion Solvers for Combinatorial Optimization. Sun & Yang. NeurIPS 2023.
>
> [2] Generation as Search Operator for Test-Time Scaling of Diffusion-based Combinatorial Optimization. Li et al. NeuIPS 2025.

---

> > ### Author Rebuttal · Reviewer_Ycfr · 2026-04-02
> >
> > Thank you for addressing my concerns. I will maintain my positive score and recommend acceptance, noting that I'm not familiar with this area of research.

---

> > > ### Author Response · Authors · 2026-04-04
> > >
> > > Thank you for maintaining your positive score. We are glad our clarifications were helpful, and will incorporate your suggestions into our revision.

---

### Official Review · Reviewer_nmAx · 2026-03-10

**Soundness:** 2
**Presentation:** 2
**Significance:** 2
**Originality:** 2
**Overall Recommendation:** 4
**Confidence:** 3

**Summary:**

The paper proposes Combinatorial Adjoint Matching (CAM) for unsupervised training of diffusion-based solvers for combinatorial optimization. The main idea is to extend adjoint matching from continuous diffusion models to discrete CO by modeling the generation process as a CTMC and introducing a binary/flip-gradient formulation that enables a discrete analogue of adjoint propagation. The method is further adapted to CO with several practical design choices, including a surrogate flip-gradient construction for general problems and clip-based training. Experiments on MIS and TSP show that CAM improves over prior unsupervised baselines and is competitive with supervised diffusion-based solvers on the tested settings.

**Compliance With Llm Reviewing Policy:**

Affirmed.

**Final Justification:**

I believe the paper still has some issues, but its strengths are still worthy of encouragement.

**Key Questions For Authors:**

1. The paper positions CAM as a systematic extension of adjoint matching to discrete combinatorial optimization, but it also discusses concurrent DAM in the related work. Could the authors clarify more precisely what they view as the main novelty beyond this concurrent line, and which parts are genuinely new in CAM?
2. Theorem 3.2 seems central to the method, but the appendix proof still contains typos, notation issues, an unresolved placeholder, and a skipped case. Could the authors provide a cleaned-up and fully self-contained argument, and clarify how strongly the practical method depends on this theorem?
3. The empirical study is currently limited to MIS and TSP, and the ablation is essentially only on the number of clips $k$. Could the authors provide more component-wise analysis? This would affect both my soundness and significance assessment.

**Limitations:**

Not fully. The limitations discussion could be strengthened.

**Strengths And Weaknesses:**

I think the paper is tackling a meaningful problem: reducing reliance on near-optimal supervised data for diffusion-based CO is worthwhile, and adapting adjoint-style training to discrete settings is an interesting direction. The empirical results are also reasonably promising, especially on MIS, where CAM is consistently better than the unsupervised baselines reported in the paper; on TSP, it is still competitive and remains close to the supervised diffusion models. I also think the paper has a clear high-level motivation, and the flip-gradient view is a sensible way to make adjoint-style ideas work in discrete spaces.

I have three main concerns. First, while the paper frames itself as a systematic extension of adjoint matching to discrete settings, it also acknowledges a concurrent Discrete Adjoint Matching work, so the high-level novelty feels more limited than the framing suggests. Second, the technical support around Theorem 3.2 is not yet in good shape. This theorem is central to the method, but the appendix proof still contains obvious typos, notation issues, an unresolved placeholder, and a skipped case, which makes the theory look not fully polished in its current form. Third, the experimental section is somewhat thin for the strength of the claims. The evaluation is limited to MIS and TSP, and the ablation is essentially only on the number of clips $k$. I would have liked to see more component-wise analysis together with a clearer discussion of sensitivity to post-processing and sampling. Overall, I see merit in the idea, but these issues are enough for me to lean weak reject.

---

> ### Author Rebuttal · Authors · 2026-03-31
>
> We want to thank the reviewer for helpful feedback. **Additional results are accessible from this [link](https://drive.google.com/file/d/1ehRcd0WI1R8mYWQi22s3rB_nXGFfCreu/view?usp=drive_link)**.
>
> Here are the detailed response:
>
> ---
>
> > it also acknowledges a concurrent Discrete Adjoint Matching work, so the high-level novelty feels more limited than the framing suggests…. Could the authors clarify more precisely what they view as the main novelty beyond this concurrent line, and which parts are genuinely new in CAM?
>
> We thank the reviewer for raising this point. While both works share the similar method name, they differ fundamentally in mechanism:
>
> - CAM performs **trajectory-wise adjoint propagation**, enabling credit assignment along a single sampled path.
> - DAM relies on state-wise **Monte Carlo estimation over multiple rollouts**, without propagating gradients through the trajectory.
>
> As a result, **CAM is a closer discrete analogue of classical adjoint methods**, while DAM follows a sampling-based optimal target estimation paradigm, deviating from the adjoint paradigm.
>
> We will clarify this distinction more precisely in the revision.
>
> ---
>
> > Second, the technical support around Theorem 3.2 is not yet in good shape. … Could the authors provide a cleaned-up and fully self-contained argument, and clarify how strongly the practical method depends on this theorem?
>
> We thank the reviewer for the suggestion. Due to space constraints, we provide a concise, self-contained argument below.
>
> **Proof sketch.**
> Assume there is a unique optimal solution $x^\star$, and the terminal cost $g_\beta(x)=g(x)/\beta$ satisfies $g(x)\propto \|x-x^\star\|_1$. Hence the optimal cost-to-go depends only on the Hamming distance to $x^\star$.
>
> Consider neighboring states $x$ and $x^{(i)}$ differing in one coordinate.
>
> - If both are non-optimal, they share at least one incorrect coordinate relative to $x^\star$. The optimal control can flip such a coordinate in both states, yielding identical control, so the difference term is zero.
>
> - If $x=x^\star$ and $x^{(i)}\neq x^\star$, then $x^{(i)}$ differs from $x^\star$ in exactly one coordinate. Since $g_\beta(x)=g(x)/\beta$, the terminal penalty diverges as $\beta\to 0$. For sufficiently small $\beta$, the optimal control concentrates on the correcting flip, making the probability of not reaching $x^\star$ before terminal time negligible. Thus the cost-to-go difference reduces to the one-bit marginal contribution, which matches the instantaneous difference, and the terms cancel. The case $x\neq x^\star$, $x^{(i)}=x^\star$ is symmetric.
>
> This establishes the claim.
>
> ---
>
> > The evaluation is limited to MIS and TSP
>
> Thank you for your suggestion. We further extend our evaluation to the capacited vehicle routing problem (CVRP) and  the MaxCut problem. Please see Section A at the above [link](https://drive.google.com/file/d/1ehRcd0WI1R8mYWQi22s3rB_nXGFfCreu/view?usp=drive_link).
>
> To summarize the results:
> - On CVRP, our proposed CAM has achieved comparable performance to the existing supervised diffusion models, and outperforms supervised flow matching and the unsupervised SDDS baselines.
> - On MaxCut, CAM even outperforms the human-designed heuristic solver.
>
> ---
>
> >  and the ablation is essentially only on the number of clips . I would have liked to see more component-wise analysis together with a clearer discussion of sensitivity to post-processing and sampling.
>
>
> Following your suggestions, we have strengthened the ablations in Section B at the same [link](https://drive.google.com/file/d/1ehRcd0WI1R8mYWQi22s3rB_nXGFfCreu/view?usp=drive_link), including
> - **The ablation on the regularization term**: Table 3 & Figure 1
>
> | MIS (ER-[700–800]) | CAM w/o regularization | CAM w/ regularization |
> |--------------------|------------------------|------------------------|
> | GAP ↓              | 18.70%                 | **1.54%**              |
>
> - **The ablation on the inference-time diffusion steps**: Figure 2
>
> - **The ablation on the different training targets** (vs. local ones, weak local search heuristics): Table 4, Figure 3&4
>
> | Setting | No adjoint | Local search w/ greedy | Standard CAM |
> |--------|------------|------------------------|----------|
> | GAP ↓ | 2.83% | 2.36% | **2.18%** |
>
> Across all settings, results consistently show the improvement of CAM and the necessity of the current CAM design.
>
> ---
>
> > Not fully. The limitations discussion could be strengthened.
>
> Thank you for the suggestion. Currently, CAM still sometimes relies on the local signals (through multiple clips), and external heuristic for discrete gradient computation. Although this is acceptable in the CO domain, future work shall explore the self-driven MCMC sampler at the terminal state, enhancing its generalization to other discrete tasks.

---

> > ### Author Rebuttal · Reviewer_nmAx · 2026-04-03
> >
> > My concerns have been adequately addressed.  I would be happy to improve my score.

---

> > > ### Author Response · Authors · 2026-04-04
> > >
> > > We sincerely thank the reviewer for the positive update. We are glad that our responses have addressed the concerns, and we will sure include the them into our paper accordingly.

---

### Official Review · Reviewer_agsh · 2026-03-11

**Soundness:** 3
**Presentation:** 3
**Significance:** 2
**Originality:** 3
**Overall Recommendation:** 4
**Confidence:** 3

**Summary:**

The paper proposes Combinatorial Adjoint Matching (CAM), an unsupervised training framework for diffusion-based CO solvers. It adapts the continuous Adjoint Matching method to discrete spaces by modeling the diffusion generation process as a continuous-time Markov chain, which is restricted to a 1-Hamming ball. To handle with the non-differentiability of discrete spaces, the authors introduce a *flip-gradient* and establish a discrete chain rule. This rule is constructed under the assumption that the cost function is proportional to the Hamming distance to the nearest optimum. To compute the flip-gradient practically, the method uses exact closed-form equations for QUBO problems (like MIS) and a surrogate objective derived from local search heuristics (like 2-opt) for non-QUBO problems (like TSP). Empirically, the method demonstrates strong sample efficiency and outperforms existing unsupervised diffusion baselines on MIS and TSP.

**Compliance With Llm Reviewing Policy:**

Affirmed.

**Final Justification:**

The authors' rebuttal clearly addresses my concerns by 1) clarifying the key component of CAM and the root of Theorem 3.2; 2) providing more experiments on multiple tasks; 3) more ablations. I believe the revised paper can be of good quality. I think the intuition behind CAM is clear to improve the training of diffusion models, and it can be flexibly combined with other methods. Thereby I raise my score to 4, and suggest the authors reconstructing the paper and clarifying the scope of CAM.

**Key Questions For Authors:**

- What if CAM is to solve a problem that does not fit the assumption in Theorem 3.2?
- Please elaborate more on the use of heuristic solver in CAM and provide ablation analysis.

**Limitations:**

The authors does not discuss the limitation is the manuscript. The authors overlook the severe limitations of their theoretical assumptions in practical NP-hard scenarios and the use of heuristic solvers.

I suggest the authors reconsider the scope of CAM. For example, consider under what situation CAM can work perfectly, like QUBO-formulated problems?

**Strengths And Weaknesses:**

## Strengths

- I think this paper is of originality. I never read a paper introducing Adjoint Matching to discrete CO. Besides my first concerned weakness, I think the derivation and the proof is elegant.
- Although some typos, the paper is well-structured and the derivations are clear. As not an expert for NCO, I can follow this paper easily.

## Weaknesses

- Unrealistic assumption in Theorem 3.2. The assumption that the terminal cost satisfies $g_{\beta}(x) \propto || x-\Pi_g(x)||_1$ need a smooth optimization landscape. However, in reality, the search spaces of NP-hard CO problems are highly non-convex, and filled with deceptive local optima. The theory and the CAM is based on this assumption, weakening the soundness.
- The use of heuristic solver for TSP hinders the contribution of CAM. The authors use greedy decoding + 2-OPT for post-processing the generated solutions of the model during training and inference. I think this is a distillation of the heuristic solver, which is not in an unsupervised manner, and it is a unfair comparison toward other unsupervised baselines.
- Limited evaluation tasks like CVRP.
- Typos:
   - line 130, "required the backpropagation over" -> "required backpropagation over";
   - line 303, "We We" -> "We";
   - line 317, "I set the" -> "We set the";
   - line 562, "thje expectation" -> "the expectation";
   - line 572, "according to ??";
   - line 579, "When neither $x$ or $x^{(i)}$" -> "When neither $x$ nor $x^{(i)}$";
   - line 580, "Since the control are" -> "Since the control is".

---

> ### Author Rebuttal · Authors · 2026-03-31
>
> We want to thank the reviewer for helpful feedback. **Additional results are accessible from this [link](https://drive.google.com/file/d/1ehRcd0WI1R8mYWQi22s3rB_nXGFfCreu/view?usp=drive_link)**.
>
> Here are the detailed response:
>
> ---
>
> > Unrealistic assumption in Theorem 3.2. The assumption that the terminal cost satisfies $g_{\beta}(x)\propto \| x-\Pi_g(x)\|_1$ need a smooth optimization landscape.  However, in reality, the search spaces of NP-hard CO problems are highly non-convex, and filled with deceptive local optima.
>
> > What if CAM is to solve a problem that does not fit the assumption in Theorem 3.2?
>
> We agree that Theorem 3.2 relies on an idealized smoothness assumption that does not strictly hold in NP-hard CO.
>
> **However, CAM does not require this assumption to hold in practice.**
>
> The theorem serves to:
> - justify the existence of a tractable adjoint signal in a simplified setting,
> - motivate the design of the flip-gradient and trajectory propagation.
>
> In practice:
> - both MIS and TSP violate this assumption,
> - yet CAM remains effective, indicating that the method **is robust to deviations from the idealized setting**.
>
> We will clarify in the revision that Theorem 3.2 is a guiding analysis, not a correctness condition.
>
> ---
>
> > The use of heuristic solvers for TSP hinders the contribution of CAM. …  I think this is a distillation of the heuristic solver, which is not in an unsupervised manner.
>
> We would like to clarify that: **CAM does not learn to imitate heuristics; it uses them only to generate improvement signals, which are then propagated across the trajectory**.
> Specifically:
>
> - A locally improved solution (via greedy + 2-opt) is used **only to define a directional signal**,
> - This signal is then **propagated backward through the trajectory** via adjoint matching.
>
> Thus, CAM learns how **intermediate decisions contribute to final improvements**, rather than replicating the heuristic itself. This is supported by our additional ablations in Section B.3 via the above [link](https://drive.google.com/file/d/1ehRcd0WI1R8mYWQi22s3rB_nXGFfCreu/view?usp=drive_link):
>
> - Direct imitation of 2-opt (no adjoint propagation) performs significantly worse,
> - CAM with weaker heuristics still outperforms imitation-based variants.
>
> | Setting | No adjoint | Local search w/ greedy | Standard CAM |
> |--------|------------|------------------------|----------|
> | GAP ↓ | 2.83% | 2.36% | **2.18%** |
>
>
> Therefore, the performance gain arises from adjoint propagation, not from the heuristic imitation.
>
> ---
>
> > and it is an unfair comparison toward other unsupervised baselines.
>
> Thank you for your question. We would like to clarify that on TSP:
>
> - All diffusion models use the same post-processing (greedy + 2-opt) at inference.
> - Our unsupervised baseline SDDS also uses 2-opt during training to evaluate rewards.
>
> Therefore, the comparison is fair and consistent across methods.
>
> ---
>
> > Limited evaluation tasks like CVRP.
>
>
> Thank you for your suggestion.  We further extend our evaluation to the capacitated vehicle routing problem (CVRP) and  the MaxCut problem. Please see Section A via the above [link](https://drive.google.com/file/d/1ehRcd0WI1R8mYWQi22s3rB_nXGFfCreu/view?usp=drive_link).
>
> To summarize the results:
> - On CVRP, our proposed CAM has achieved comparable performance to the existing supervised diffusion models, and outperforms supervised flow matching and the unsupervised SDDS baselines.
> - On MaxCut, CAM even outperforms the human-designed heuristic solver.
>
> ---
>
> > I suggest the authors reconsider the scope of CAM. For example, consider under what situation CAM can work perfectly, like QUBO-formulated problems?
>
> Thank you for the suggestion. Existing unsupervised diffusion models, such as DiffUCO and RLNN, are primarily limited to QUBO. **CAM represents a first step toward extending this paradigm to more general CO problems**. We believe its comparable performance to supervised diffusion models highlights its potential and provides a useful direction for future research in this area.
>
> Regarding the reliance on heuristic solvers, we would like to clarify that **heuristic post-processing is standard across existing neural CO solvers and is not a limitation specific to CAM**. But we do acknowledge that there is room for improvement in this component. A promising direction is to further integrate neural components to enhance the quality of these heuristics (e.g., neural-guided refinement), which could make CAM more robust even when using weaker heuristic operators.

---

> > ### Author Rebuttal · Reviewer_agsh · 2026-04-03
> >
> > Thanks for the detailed rebuttal. I am considering adjust my ratings. During checking the authors' response and details in related works, I have the following remaining concerns:
> > 1. Existing diffusion-based NCO methods seem not only equip the respective methods with 2-OPT, but also compare related diffusion methods in a heuristic-free setting to check the corresponding precision. I suggest adding this discussion.
> > 2. It seems that two more recent diffusion-based baselines [1-2].
> > 3. "*A promising direction is to further integrate neural components to enhance the quality of these heuristics (e.g., neural-guided refinement), which could make CAM more robust even when using weaker heuristic operators.*" It seems that [3] has done the similar idea. Can you elaborate more on this?
> >
> > I'm deciding whether to increase to 3 or 4. I would be more than happy if you can address these questions.
> >
> > [1] Fast T2T: Optimization Consistency Speeds Up Diffusion-Based Training-to-Testing Solving for Combinatorial Optimization. NeurIPS 2024.
> >
> > [2] Generation as search operator for test-time scaling of diffusion-based combinatorial optimization. NeurIPS 2025.
> >
> > [3] Neural Combinatorial Optimization with Heavy Decoder. NeurIPS 2023.

---

> > > ### Author Response · Authors · 2026-04-04
> > >
> > > Thank you for your positive feedback and follow-up questions. Here we detail the response to each of your remaining concerns.
> > >
> > > ### **1. Comparison in heuristic-free setting**
> > >
> > > Thank you for the suggestion. Here we report the results **without 2-OPT**. As expected, the absolute gaps increase due to the absence of local refinement, but **the relative ranking across methods remains consistent** (CAM is still the second best on TSP-500 and the best on TSP-1000), indicating that the performance gain of CAM does not hold only after post-processing.
> > >
> > > | Dataset | CAM | DIFUSCO | FM | SDDS |
> > > |-----------|----------|----------|----------|----------|
> > > | TSP-500 | 12.93% | **11.18%** | 14.38% | 15.29% |
> > > | TSP-1000 | **13.49%** | 16.95% | 13.80% | 17.13% |
> > >
> > > ---
> > >
> > > ### **2. Recent diffusion-based baselines (Fast T2T, GenSCO)**
> > >
> > > Thank you for pointing out these recent works. We have discussed them in Section 6.2 and further clarify their relation here.
> > >
> > > These methods introduce improvements beyond standard diffusion solvers, such as:
> > > - **Fast T2T**: incorporating consistency between training and test-time objectives, together with gradient-based refinement
> > > - **GenSCO**: treating diffusion sampling as a **search operator** within an iterative test-time optimization loop
> > >
> > > Importantly, these approaches are **orthogonal to our contribution**, that is: our method improves the **training of diffusion models**, while these methods primarily enhance **inference-time procedures**, such as guidance, search, or refinement.
> > >
> > > In these frameworks, the diffusion model serves as a **core generative backbone embedded within a larger solving pipeline**. Therefore, CAM can be naturally combined with these approaches, for example, using CAM as the base model within GenSCO-style test-time search, suggesting a promising direction for future work.
> > >
> > > ---
> > >
> > > ### **3. Relation to neural-augmented search operators ([3])**
> > >
> > > Thank you for pointing out this connection.
> > >
> > > Method [3] (Heavy Decoder) follows a **neural-guided local search paradigm**, where a model trained with supervision from high-quality solutions is used at inference time to guide neighborhood reconstruction (e.g., destroy-and-repair). In this setting, **the neural network has already learned how to improve solutions from near-optimal data**.
> > >
> > > In contrast, our setting is fundamentally different: we do not assume access to optimal or near-optimal solutions, and instead aim to **estimate improvement signals/gradient from a sub-optimal model**.
> > >
> > > Our approach can be interpreted as operating over a **sequence of progressively refined distributions**. Starting from samples generated by the current policy (a sub-optimal distribution $p_i$), we construct a reward-tilted distribution:
> > >
> > > $$p_{i+1}(x) \propto p_i(x)\exp(-g(x)/\tau)$$
> > >
> > > which assigns higher probability to better solutions.
> > >
> > > In practice, we could:
> > > - sample candidate solutions using the current policy (e.g., via a stochastic / MCMC-style sampler), each decoded via the weak heuristic
> > > - extract **relative improvement signals** within the sampled set,
> > > - and use them to estimate the score of $p_{i+1}$ as the surrogate flip gradient
> > >
> > > After enough iterations, the induced distribution becomes increasingly concentrated on higher-quality solutions, and the surrogate flip gradient becomes more accurate as well.
> > >
> > > This setting is inherently more challenging, as the model must bootstrap improvement signals without access to near-optimal supervision.
> > >
> > > ---
> > >
> > > We hope these clarifications addresses the your additional concerns.

---

### Official Review · Reviewer_ynFm · 2026-03-13

**Soundness:** 3
**Presentation:** 2
**Significance:** 2
**Originality:** 2
**Overall Recommendation:** 3
**Confidence:** 2

**Summary:**

The paper addresses the issue of tackling Combinatorial Optimization through AI methods. It develops an unsupervised method for discrete problems, building on recent approaches in the continuous counterpart of the same problem. A practical algorithm is presented, theoretically analysed, and tested on the Travel Salesman Problem and the Maximum Independent Set problem.

**Compliance With Llm Reviewing Policy:**

Affirmed.

**Ethical Review Concerns:**

No ethical concerns

**Key Questions For Authors:**

Could you please provide further justification and explanation for the theoretical framework adaptations proposed in Section 4.1? Does your prior theoretical analysis still apply under these adaptations? Is there a clear motivation for CAM's improved performance in TSP compared with the other considered unsupervised methods? The differences in the effect of the number of clips are related to the balance between the global and local scales studied with the clips. Have you also considered the effect of the clips in the approximation of the discrete gradient in the first place?

**Limitations:**

A better explanation of the work's limitations is needed. In particular, more focus is needed on defining the applicability boundaries of the methodology, clarifying which further studies are needed, and identifying which practical applications necessitate an unsupervised approach like the one considered in the paper, given that supervised and classical strategies are not feasible.

**Strengths And Weaknesses:**

Strengths: The presented method overcomes previous issues in tackling discrete problems in combinatorial optimization by introducing a suitable definition of the discrete gradient and by properly adapting recent continuous-setting frameworks.


Weaknesses: The experimentally validated algorithm is not fully supported by the developed theory. In particular, several approximations are needed to make the model trainable. An example is provided by replacing the regularisation term in (37) with that in (40). On the numerical side, the methods proposed for computing the discrete gradient seem ineffective, and the analysis of these limitations is too heuristic. The discussion of the experiments needs to be expanded and investigated in more detail. For instance, the reasons for which the proposed model works well in the TSP despite being unsupervised seem not clearly motivated (in particular with respect to the other UL methods). Some more minor comments are: In Proposition 3.1 $\beta$ is not specified. Throughout, some typos need to be fixed, such as $0,1^n$ instead of {0,1}$^n$ in (16), "sufficiently $\beta$" in line 193, or "We We" in line 303.

---

> ### Author Rebuttal · Authors · 2026-03-31
>
> We want to thank the reviewer for helpful feedback. **Additional results are accessible from this [link](https://drive.google.com/file/d/1ehRcd0WI1R8mYWQi22s3rB_nXGFfCreu/view?usp=drive_link)**.
>
> Here are the detailed response.
>
> ---
>
> > several approximations are needed to make the model trainable.
>
> We thank the reviewer for highlighting the role of approximations. **Importantly, CAM does not rely on these approximations for correctness, but only for practical optimization in non-convex discrete spaces.**
>
> Our method separates:
> - an idealized adjoint framework that clarifies how terminal objective information should propagate, and
> - practical instantiations that addresses the non-convexity or approximates this propagation under discrete constraints.
>
> The core mechanism, **trajectory-wise credit assignment via adjoint propagation**, remains valid regardless of these approximations.
>
> ---
> > Could you please provide further justification and explanation for the theoretical framework adaptations proposed in Section 4.1? Does your prior theoretical analysis still apply under these adaptations?
>
> > An example is provided by replacing the regularization term in (37) with that in (40).
>
> We thank the reviewer for this question. **The adaptations in Section 4.1 are designed to improve optimization in non-convex discrete landscapes, rather than to replace the underlying theoretical mechanism.**
>
> - Eq. (37) **computes the gradient** signal aligned with the original objective in Eq. (23). In an idealized setting, this provides a valid descent direction.
>
> - Eq. (40) is introduced to **prevent the dynamics from getting trapped in poor local optima.** Its role is analogous to stabilization mechanisms in SGD/Adam: it does not change the objective or gradient direction, but avoids vanishing gradients. This design is motivated by prior work [1,2], where **similar regularization is shown to be necessary** in combinatorial optimization.
>
> - The design in Eq. (40) **does not affect the correctness of the gradient computation** in Eq. (30)
>
> We will clarify in the revision that Eq. (37) provides the objective-aligned gradient signal, while Eq. (40) is an enhancement for non-convex optimization rather than a change to the underlying objective.
>
> [1] Revisiting Sampling for Combinatorial Optimization. Sun et al. ICML 2022.
>
> [2] Regularized Langevin Dynamics for Combinatorial Optimization. Feng & Yang. ICML 2025.
>
> ---
>
> > On the numerical side, the methods proposed for computing the discrete gradient seem ineffective, and the analysis of these limitations is too heuristic.
>
> We clarify that the role of the discrete gradient in CAM is **not to provide an exact descent direction**, but to supply a consistent improvement signal that can be propagated through the trajectory.
>
> - For QUBO problems, this signal is exact.
> - For general CO (e.g., TSP), the surrogate gradient provides a locally valid improvement direction, which is sufficient for adjoint propagation.
>
> **The key requirement is directional consistency, not exactness**, which is why CAM remains effective even with approximate gradients.
>
> ---
>
> > The discussion of the experiments needs to be expanded and investigated in more detail. For instance, the reasons for which the proposed model works well in the TSP despite being unsupervised seem not clearly motivated (in particular with respect to the other UL methods).
>
> The key reason CAM performs well on TSP despite being unsupervised is that **it propagates discrete gradients along entire solution trajectories via adjoint matching**, rather than relying on local or sample-based learning.
>
> Prior unsupervised diffusion methods face two limitations:
> - **Local improvement learning** (e.g., RLNN) relies on step-wise objectives that are inconsistent across the trajectory and fail to enforce global structure.
> - **Self-sampled trajectory learning** (e.g., SDDS) depends on discovering high-quality solutions through sampling, which leads to sparse and unstable training signals in high-dimensional spaces.
>
> CAM addresses both issues by using discrete gradient (adjoint) propagation to assign credit to intermediate states along the trajectory, enabling consistent and informative updates. This is supported by our additional ablations in Section B.3 in the above [link](https://drive.google.com/file/d/1ehRcd0WI1R8mYWQi22s3rB_nXGFfCreu/view?usp=drive_link):
> - **Removing adjoint propagation** (no adjoint) significantly degrades performance as below (also Table 4, Figure 3 in the above link), showing that local improvement alone is insufficient.
>
> | TSP-500 | No adjoint | Standard CAM |
> |--------|-----------|----------|
> | GAP ↓ | 2.83% |  **2.18%** |
>
> - **Training with self-sampled trajectories** (SDDS) exhibits unstable and divergent dynamics (Figure 4), confirming the difficulty of learning from sparse rewards.
>
> These results demonstrate that CAM’s advantage comes from **global consistency and informative updates**.

---

> > ### Author Rebuttal · Reviewer_ynFm · 2026-04-03
> >
> > Thank you for the additional details. Looking at the authors' responses, a major concern still persists. Since the developed theory and the numerical experiments diverge, could you please clarify which properties of the theoretical analysis remain valid in the empirical setting?

---

> > > ### Author Response · Authors · 2026-04-04
> > >
> > > Thank you for this important question. We clarify that the **properties of the adjoint state remain fully valid in the empirical setting**, and that the theoretical analysis and practical optimization play two distinct roles in our method.
> > >
> > > Specifically, the theoretical framework defines **how the training signal is computed**, while the practical adaptations determine **how this signal is used for optimization**.
> > >
> > > First, the adjoint state is a **well-defined object derived from the stochastic control formulation (Eq. (23))**, analogous to how gradients are defined by an objective function. It depends only on the cost functional and transition dynamics, and is independent of how the control is parameterized or optimized in practice. Therefore, its structural properties, such as the terminal condition (Eq. (27)) and recursion (Eq. (29)), remain unchanged.
> > >
> > > The practical modifications (e.g., discretization, regularization, and surrogate gradients) affect **how this signal is used to update the model**, but do not alter the signal itself. As a result, while the optimization procedure is adapted for stability in non-convex discrete spaces, the **core theoretical mechanism, propagating objective-aligned information through the adjoint, remains intact**.
> > >
> > > We hope this clarification addresses the your concern.

---

### Decision · Program_Chairs · 2026-04-30

**Decision:**

Accept (regular)

**Comment:**

This paper develops an neural solver for combinatoric optimization problem via generalizing adjoint matching to discrete combinatorial domains. This is an interesting idea because prior diffusion approaches to neural solver often requires large collections of near-optimal solutions, which presents a critical bottleneck for their scalability and generalization. The paper addresses this by extending adjoint matching, unsupervised diffusion training framework based on chain-rule–style gradient propagation in continuous spaces, to discrete combinatorial domains. The resulting algorithm is theoretically analyzed and tested on the Traveling Salesman and Maximum Independent Set problems.

Overall, the reviewing panel largely appreciate the contribution of this paper. The empirical strength of this paper is recognized. Its theoretical analysis might have some limitations & the experimentally validated algorithm is not fully supported by the developed theory as it seems to require a lot of approximations. This appears to be the key concern that remains. I have read the authors' rebuttal follow-up which, I think, does not fully address this point. However, I view this limitation as future research scope rather that one that would severely hamper the existing practical contribution. Accordingly, I am happy to recommend weak accept for this paper.